# Whitened Score Diffusion:
# A Structured Prior for Imaging Inverse Problems

**Jeffrey Alido[1]**    **Tongyu Li[1]**    **Yu Sun[2]**    **Lei Tian[1]**

[1]Department of Electrical and Computer Engineering,
Boston University, Boston, MA 02215, USA
[2]Department of Electrical and Computer Engineering,
Johns Hopkins University, Baltimore, MD 21205, USA
`{jalido, tongyuli, leitian}@bu.edu, ysun214@jh.edu`

## Abstract

Conventional score-based diffusion models (DMs) may struggle with anisotropic Gaussian diffusion processes due to the required inversion of covariance matrices in the denoising score matching training objective [58]. We propose Whitened Score (WS) diffusion models, a novel framework based on stochastic differential equations that learns the Whitened Score function instead of the standard score. This approach circumvents covariance inversion, extending score-based DMs by enabling stable training of DMs on arbitrary Gaussian forward noising processes. WS DMs establish equivalence with flow matching for arbitrary Gaussian noise, allow for tailored spectral inductive biases, and provide strong Bayesian priors for imaging inverse problems with structured noise. We experiment with a variety of computational imaging tasks using the CIFAR, CelebA ($64 \times 64$), and CelebA-HQ ($256 \times 256$) datasets and demonstrate that WS diffusion priors trained on anisotropic Gaussian noising processes consistently outperform conventional diffusion priors based on isotropic Gaussian noise. Our code is open-sourced at `github.com/jeffreyalido/wsdiffusion`.

## 1 Introduction

Diffusion models (DMs) are a powerful class of generative models that implicitly learn a complex data distribution by modeling the (Stein) score function [48, 49, 18, 13, 26, 25]. The score function is then plugged into a reverse denoising process described by an ordinary differential equation (ODE) or a stochastic differential equation (SDE) to generate novel samples from noise. Typically, the forward noising process is defined by adding different levels of isotropic Gaussian noise to a clean data sample, which enables a simple and tractable denoising score matching (DSM) objective [58]. However, the DSM objective exhibits instability when the forward diffusion noise covariance is ill-conditioned or singular, as its computation requires inverting the covariance matrix.

Flow matching (FM) [33, 35, 1, 65] is an alternative generative modeling paradigm that reshapes an arbitrary known noise distribution into a complex data distribution according to an implicit probability path constructed by the flow. For the isotropic Gaussian case, [34, 52] established that FM and DMs are equivalent up to a rescaling of the noise parameters that define the SDE and probability paths. However, for anisotropic Gaussian noise, there exists a gap between score-based DMs and FM, where score-based DMs cannot be as easily trained for arbitrary Gaussian forward noising processes due to the necessary inversion of the covariance matrix in the conditional score [58].

A denoising DM capable of denoising structured, correlated noise is desirable in many scientific inverse problems, especially in imaging, as it may serve as a rich Bayesian prior [49, 16, 63, 28]. Imaging through fog, turbulence and scattering [3, 64, 31], wide-field microscopy [39], diffraction tomography [32, 30], optical coherence tomography (OCT) [21], interferometry [56] and many other

39th Conference on Neural Information Processing Systems (NeurIPS 2025).

imaging modalities have an image formation process corrupted by structured, spatially correlated noise [62, 6], in contrast to the widely assumed additive isotropic (white) Gaussian noise. Conventional DMs are trained on isotropic Gaussian noise, which may render them practically insufficient Bayesian priors for realistic use cases with correlated noise.

Motivated by FM's ability to model arbitrary probability paths and the expressiveness of diffusion priors for inverse problems, we propose **Whitened Score Diffusion**, a framework for learning DMs based on arbitrary Gaussian noising processes. Instead of learning the (time-dependent) score function, $\nabla_{\mathbf{x}_t} \log p_t(\mathbf{x}_t)$, we learn $\mathbf{G}_t \mathbf{G}_t^\top \nabla_{\mathbf{x}_t} \log p_t(\mathbf{x}_t)$, with $\mathbf{G}_t$ the diffusion matrix in the forward diffusion process (Fig. 1). We term our framework Whitened Score (WS) DMs, after the whitening transformation that transforms the score vector field into an isotropic vector field. This extends the current SDE framework for score-based DMs as it avoids the computation of the inverse covariance for any anisotropic Gaussian noise in DSM objective, enabling an arbitrary choice of Gaussian probability paths, similar to FM. We elaborate on the equivalence of our framework to FM and draw a connection to the reverse-time diffusion process derivation by [8], where $\mathbf{G}_t \mathbf{G}_t^\top \nabla_{\mathbf{x}_t} \log p_t(\mathbf{x}_t)$ is a *predictable process* of the stochastic term in a reverse-time SDE.

This work presents an extension of score-based DMs to arbitrary Gaussian forward processes, bridging a gap between DMs and FM. Our framework enables a principled construction of denoising generative priors that incorporate spectral bias aligned with correlated measurement noise, leading to improved performance in inverse problems with structured noise. Empirical results on CIFAR-10, CelebA ($64 \times 64$), and CelebA-HQ ($256 \times 256$) across several imaging tasks show consistently higher peak signal-to-noise ratio (PSNR) reconstructions compared to conventional models trained with isotropic noise. Our contributions are: (i) a framework for training DMs that supports arbitrary Gaussian probability paths, (ii) theoretical insights and a connection to FM, and (iii) a demonstration of effective priors for imaging inverse problems under structured noise.

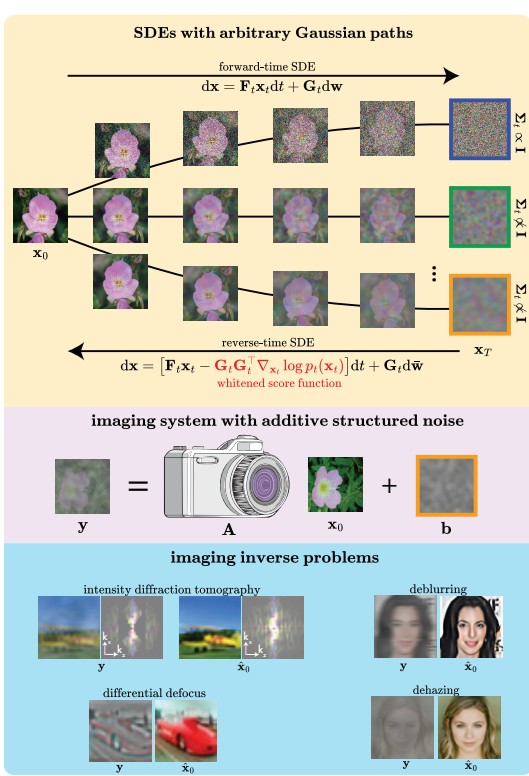

Figure 1: Our framework enables arbitrary Gaussian diffusion processes, allowing us to train a denoising DM on a diverse set of structured noise. The WS DM applies to a variety of imaging inverse problems corrupted with correlated, structured noise.

## 2 Background

### 2.1 Score-based diffusion models

Score-based DMs are a class of generative models that estimate a probability density function $p(\mathbf{x}_0)$ by reversing a time-dependent noising process. In continuous time, the forward noising process is described by an Itô SDE in the form of,

$$\mathrm{d}\mathbf{x}_t = \mathbf{F}_t \mathbf{x}_t \mathrm{d}t + \mathbf{G}_t \mathrm{d}\mathbf{w}. \tag{1}$$

$\mathbf{F}_t \in \mathbb{R}^{m \times m}$ is the drift coefficient, $\mathbf{w} \in \mathbb{R}^m$ is the standard Wiener process (Brownian motion), and $\mathbf{G}_t \in \mathbb{R}^{m \times m}$ is the diffusion matrix that controls the structure of the noise. The noise level is indexed by time $t \in [0, T]$ such that $\mathbf{x}_0 \sim p(\mathbf{x}_0)$, $\mathbf{x}_T \sim \mathcal{N}(\mathbf{0}, \mathbf{I})$ and $\mathbf{x}_t \sim p(\mathbf{x}_t \mid \mathbf{x}_0)$, a probability transition Gaussian kernel defined by Eq. 1.

The corresponding reverse-time SDE for Eq. 1 is:

$$d\mathbf{x}_t = \left[\mathbf{F}_t\mathbf{x}_t - \mathbf{G}_t\mathbf{G}_t^\top \nabla_{\mathbf{x}_t} \log p_t(\mathbf{x}_t)\right] dt + \mathbf{G}_t \, d\bar{\mathbf{w}}_t, \tag{2a}$$

and the deterministic ODE, also known as probability flow, with the same time-marginals is:

$$d\mathbf{x}_t = \left[\mathbf{F}_t\mathbf{x}_t - \frac{1}{2}\mathbf{G}_t\mathbf{G}_t^\top \nabla_{\mathbf{x}_t} \log p_t(\mathbf{x}_t)\right] dt, \tag{2b}$$

where $\bar{\mathbf{w}}$ is the reverse-time standard Wiener process and $\nabla_{\mathbf{x}_t} \log p_t(\mathbf{x}_t)$ is the Stein score function.

Sampling from $p(\mathbf{x}_0)$ requires solving Eq. 2 and thereby knowing the score function, $\nabla_{\mathbf{x}_t} \log p_t(\mathbf{x}_t)$. [48, 49] approximated the score function with a neural network $\mathbf{s}_\theta(\mathbf{x}_t, t)$ by optimizing the DSM objective [58]:

$$\hat{\theta} = \arg\min_\theta \mathbb{E}_{t\sim U(0,1],\mathbf{x}_t\sim p(\mathbf{x}_t|\mathbf{x}_0),\mathbf{x}_0\sim p(\mathbf{x})} \left\{\|\mathbf{s}_\theta(\mathbf{x}_t, t) - \nabla_{\mathbf{x}_t} \log p_t(\mathbf{x}_t \mid \mathbf{x}_0)\|_2^2\right\}, \tag{3}$$

where the conditional score function has a closed form expression given by

$$\nabla_{\mathbf{x}_t} \log p_t(\mathbf{x}_t \mid \mathbf{x}_0) = \boldsymbol{\Sigma}_t^{-1}(\boldsymbol{\mu}_t - \mathbf{x}_t), \tag{4}$$

with $\boldsymbol{\mu}_t$ and $\boldsymbol{\Sigma}_t$ the mean and covariance of the Gaussian transition kernel $p(\mathbf{x}_t \mid \mathbf{x}_0)$. $\boldsymbol{\mu}_t$ and $\boldsymbol{\Sigma}_t$ are functions of the drift coefficient and diffusion matrix in Eq. 1 attained by solving the ODEs in Eqs. 5.50 and 5.51 in [54], creating a linearly proportional relationship as $\boldsymbol{\mu}_t \propto \mathbf{x}_0$ and $\boldsymbol{\Sigma}_t \propto \mathbf{G}_t\mathbf{G}_t^\top$. This leads the transformed score function term, $\mathbf{G}_t\mathbf{G}_t^\top \nabla_{\mathbf{x}_t} \log p_t(\mathbf{x}_t)$ in Eq. 2b to always be isotropic, as the covariance will multiply with its inverse, regardless of the diffusion coefficient, $\mathbf{G}_t$.

## 2.2 Structured Forward Processes in Diffusion Models

Conventional score-based diffusion models (DMs) typically employ uncorrelated white Gaussian noise, corresponding to a diagonal diffusion matrix $\mathbf{G}_t$ [49]. However, this formulation constrains the learned score function $\nabla_{\mathbf{x}_t} \log p(\mathbf{x}_t)$ to isotropic noise settings. Extending beyond diagonal $\mathbf{G}_t$ poses numerical challenges, as the inversion of the covariance matrix in Eq. 4 can become unstable for ill-conditioned or singular cases.

Recent studies have demonstrated that controlling the spatial frequency content of the forward noise can influence the model's inductive spectral bias and enhance generative flexibility [23]. Yet, a unified framework for training diffusion models under arbitrary noising processes remains lacking. Our work addresses this gap through the lens of score-based DMs in the SDE formulation, establishing a principled foundation for frequency-controlled and structured forward processes that can be adapted to diverse DM tasks.

Several prior approaches have introduced structured forward processes to improve generative expressivity. CLD [15] and PSLD [40] extend the state space by incorporating velocity variables, injecting noise in phase space to simplify score estimation, albeit at the cost of auxiliary dynamics. MDMs [45] and Blurring Diffusion [19] employ anisotropic or spatially correlated noise but require inversion of dense covariance matrices. Flexible Diffusion [17] parameterizes the forward SDE to allow adaptive noise scheduling, increasing model complexity and training cost.

These advances collectively underscore the importance of moving beyond isotropic Gaussian noise while revealing practical limitations related to stability and computational overhead. Motivated by this, our proposed WS model enables arbitrary Gaussian forward processes without covariance inversion, offering a simple, stable, and general mechanism for structured generative modeling.

## 2.3 Flow matching

Flow matching (FM) [33, 35, 1] is another paradigm in generative modeling that connects a noise distribution and a data distribution with an ODE

$$\frac{d\phi_t(\mathbf{x}_t)}{dt} = \mathbf{u}_t(\phi_t(\mathbf{x}_t)), \tag{5}$$

for FM vector field $\mathbf{u}_t(\mathbf{x}_t)$ and initial condition $\phi_0(\mathbf{x}_0) = \mathbf{x}_0$. Noise samples are transformed along time into a sample from the data distribution using a neural network that models the conditional FM vector field

$$\mathbf{u}_t(\mathbf{x}_t \mid \mathbf{x}_0) = \boldsymbol{\Sigma}_t'(\mathbf{x}_0)\boldsymbol{\Sigma}_t^{-1}(\mathbf{x}_0)(\mathbf{x}_t - \boldsymbol{\mu}_t(\mathbf{x}_0)) + \boldsymbol{\mu}_t'(\mathbf{x}_0), \tag{6}$$

where $\boldsymbol{\mu}_t(\mathbf{x}_0)$ and $\boldsymbol{\Sigma}_t(\mathbf{x}_0)$ are the mean and covariance of the probability path $p_t$, and $f'$ denotes the time derivative of $f$. Because $\boldsymbol{\Sigma}'_t$ is proportional to $\boldsymbol{\Sigma}_t$ up to a scalar coefficient, multiplying by the inverse to yield identity [54], the functional form of $\mathbf{u}_t(\mathbf{x}_t \mid \mathbf{x}_0)$ allows simple and stable training of FM models with arbitrary Gaussian probability paths. This diagonal matrix-yielding multiplication currently lacks in score-based models due to the necessary inversion of the covariance matrix in Eq. 4.

We note that WS aligns with FM in the sense that both frameworks aim to enable arbitrary probability paths. In Section 3.2, we present a formal connection between WS and FM. Nevertheless, FM may require new approaches to incorporate the measurement likelihood for solving inverse problems [65], whereas our WS framework can be readily combined with existing techniques for enforcing measurement consistency (see Section 2.4 for a review).

## 2.4 Imaging inverse problems with diffusion model priors

Reconstructing an unknown signal $\mathbf{x}_0 \in \mathbb{R}^m$ from a measurement $\mathbf{y} \in \mathbb{R}^n$ given a known forward model $\mathbf{y} \sim \mathcal{N}(\mathbf{A}\mathbf{x}_0, \boldsymbol{\Sigma}_{\mathbf{y}})$—with $\boldsymbol{\Sigma}_{\mathbf{y}} \in \mathbb{R}^{n \times n}$ the covariance of the additive Gaussian noise and $\mathbf{A} \in \mathbb{R}^{m \times n}$ the measurement forward model—is a central challenge in computational imaging and scientific problems. Recent advances employ DMs as flexible priors [14], using plug-and-play schemes [66, 67, 60], likelihood-guided sampling via posterior score approximations [22, 49, 9, 11, 47, 27], Markov Chain Monte Carlo (MCMC) techniques [38, 7, 61, 63, 53, 55], variational methods [16, 37], and latent DM frameworks [43, 12, 46].

Here, we adopt methods that approximate the posterior. The posterior score can be factored into the prior score and the likelihood score using Bayes's rule to arrive at a modification of Eq. 2 for the stochastic reverse diffusion

$$\mathrm{d}\mathbf{x}_t = \left[ \mathbf{F}_t \mathbf{x}_t - \mathbf{G}_t \mathbf{G}_t^\top \left( \nabla_{\mathbf{x}_t} \log p_t(\mathbf{x}_t) + \nabla_{\mathbf{x}_t} \log p_t(\mathbf{y} \mid \mathbf{x}_t) \right) \right] \mathrm{d}t + \mathbf{G}_t \, \mathrm{d}\bar{\mathbf{w}}_t, \tag{7a}$$

and the deterministic reverse diffusion

$$\mathrm{d}\mathbf{x}_t = \left[ \mathbf{F}_t \mathbf{x}_t - \frac{1}{2} \mathbf{G}_t \mathbf{G}_t^\top \left( \nabla_{\mathbf{x}_t} \log p_t(\mathbf{x}_t) + \nabla_{\mathbf{x}_t} \log p_t(\mathbf{y} \mid \mathbf{x}_t) \right) \right] \mathrm{d}t. \tag{7b}$$

The prior score is approximated by the denoising DM. However, the measurement likelihood score is intractable due to the time-dependence. Methods in [11, 5, 47, 50] make simplifying assumptions about the prior distribution, while those in [22, 10, 28, 43] treat the likelihood score approximation as an empirically designed update using the measurement as a guiding signal. All these likelihood score approximations can thus be plugged into Eq. 7 to solve the inverse problem.

A major gap in current research on imaging inverse problems is the consideration of additive *structured* noise. Most research on DM priors for imaging inverse problems has largely focused on scenarios with isotropic Gaussian noise, employing corresponding isotropic Gaussian denoising DMs. Recent work by [20] explored structured priors for imaging inverse problems using stochastic restoration priors achieving superior performance over conventional denoising DMs trained on isotropic Gaussian noising processes in cases involving both correlated and uncorrelated noise. However, a formal treatment of structured noise in diffusion-based frameworks lacks, which we seek to address.

# 3 Whitened Score Diffusion

We define our forward-time SDE with non-diagonal diffusion matrix as,

$$\mathrm{d}\mathbf{x}_t = \underbrace{-\frac{1}{2}\beta_t \, \mathbf{x}_t \mathrm{d}t}_{:=\mathbf{F}_t} + \underbrace{\sqrt{\beta_t}\mathbf{K}}_{:=\mathbf{G}_t} \, \mathrm{d}\mathbf{w}, \tag{8}$$

adopting from the variance-preserving (VP) SDE [49]. In our experiments, we constrain $\mathbf{K}$ to be in the class of circulant convolution matrices due to their ability to be implemented with the fast Fourier transform (FFT). However, our method generalizes to any $\mathbf{K}$ that is positive semidefinite. When $\mathbf{K} = \mathbf{I}$, we recover exactly the VP-SDE. The corresponding probability transition kernel of Eq. 8 is[1]

$$p(\mathbf{x}_t \mid \mathbf{x}_0) = \mathcal{N}\left( \mathbf{x}_t \mid \alpha_t \mathbf{x}_0, (1 - \alpha_t^2)\mathbf{K}\mathbf{K}^\top \right), \tag{9}$$

---

[1] The mean and covariance of the transition kernel are solved in Eqs. 5.50 and 5.51 in [54].

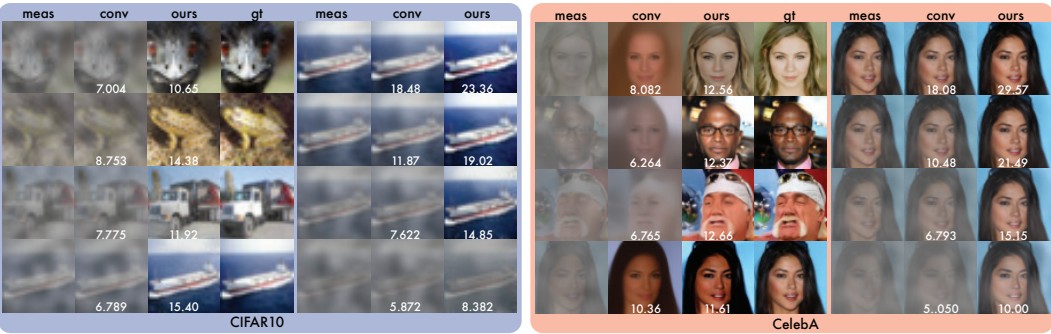

Figure 2: Denoising correlated noise on CIFAR10 and CelebA ($64 \times 64$). We benchmark our WS DM trained on anisotropic Gaussian noise with the conventional DM (conv) trained on isotropic Gaussian noise. Left: results with a fixed SNR of 0.26; Right: measurements $\mathbf{y}$ with decreasing SNR from 1.4 to 0.12 using additive grayscale noise filtered by a Gaussian kernel of std 2.5 and 5 pixels for CIFAR10 and CelebA, respectively. The PSNR is labeled in white.

where $\alpha_t = e^{-\frac{1}{2}\int_0^t \beta_s \mathrm{d}s}$. In general, $\alpha_t$ is defined as the integral of the drift coefficient from 0 to $t$, $\alpha_t = \int_0^t \mathbf{F}_s \mathrm{d}s$. By leveraging the parameterization trick for Gaussian distributions, we may rewrite Eq. 9 as the following continuous time system:

$$\mathbf{x}_t = \alpha_t \mathbf{x}_0 + \sqrt{1 - \alpha_t^2}\mathbf{K}\mathbf{z}, \quad \mathbf{z} \sim \mathcal{N}(\mathbf{0}, \mathbf{I}). \tag{10}$$

Note that we may use other drift and diffusion matrices, such as the variance-exploding (VE) SDE, ending up with scalar multiples of $\mathbf{x}_0$ and $\mathbf{G}_t\mathbf{G}_t^\top$ for the mean and covariance, respectively, given the initial conditions of $\boldsymbol{\mu}_0 = \mathbf{x}_0$ and $\boldsymbol{\Sigma}_0 = \mathbf{0}$. Specific to our SDE in Eq. 8, we define the signal-to-noise ratio (SNR) to be the ratio $\alpha_t/\sqrt{1 - \alpha_t^2}$.

### 3.1 Whitened Score matching objective

From Eq. 9, the conditional score to solve Eq. 3 is

$$\nabla_{\mathbf{x}_t} \log p(\mathbf{x}_t \mid \mathbf{x}_0) = \left((1 - \alpha_t^2)\mathbf{K}\mathbf{K}^\top\right)^{-1}\left(\alpha_t \mathbf{x}_0 - \mathbf{x}_t\right), \tag{11}$$

and inverting the matrix may often lead to instability in the score computation. For example, the condition number of a Gaussian convolution matrix grows as the Gaussian kernel $\mathbf{K}$ widens, amplifying high spatial frequency features, leading to poor model training for the DSM objective in Eq. 3.

To mitigate these numerical instabilities in the score computation during training, we apply a *whitening* transformation to the score by naturally multiplying it with $\mathbf{G}_t\mathbf{G}_t^\top$, where $\mathbf{G}_t\mathbf{G}_t^\top \propto \boldsymbol{\Sigma}_t$, the forward diffusion process covariance. Similar to DSM, for our SDE in Eq. 8, we approximate $\mathbf{G}_t\mathbf{G}_t^\top\nabla_{\mathbf{x}_t} \log p_t(\mathbf{x}_t)$ as $\mathbf{G}_t\mathbf{G}_t^\top\nabla_{\mathbf{x}_t} \log p(\mathbf{x}_t \mid \mathbf{x}_0)$ which has the following closed-form expression after canceling $\boldsymbol{\Sigma}_t$ with $\mathbf{G}_t\mathbf{G}_t^\top$:

$$\mathbf{G}_t\mathbf{G}_t^\top\nabla_{\mathbf{x}_t} \log p(\mathbf{x}_t \mid \mathbf{x}_0) = \beta_t \frac{\alpha_t \mathbf{x}_0 - \mathbf{x}_t}{1 - \alpha_t^2}. \tag{12}$$

We train a model $\mathbf{n}_\theta(\mathbf{x}_t, t)$ using the following denoising WS matching loss:

$$L = \mathbb{E}_{t\sim U(0,1), \mathbf{x}_t\sim p(\mathbf{x}_t|\mathbf{x}_0), \mathbf{x}_0\sim p(\mathbf{x})} \left\{\|\mathbf{n}_\theta(\mathbf{x}_t, t) - \mathbf{G}_t\mathbf{G}_t^\top\nabla_{\mathbf{x}_t} \log p_t(\mathbf{x}_t \mid \mathbf{x}_0)\|_2^2\right\}, \tag{13}$$

with proof in Appendix A. This objective accounts for varying levels of spatial correlation in the noise to enable our model to denoise arbitrary Gaussian noise.

This objective defines a new learning target within the broader landscape of diffusion model losses. Our approach can be seen as a generalization of noise prediction [18] to the setting of correlated Gaussian noise, where the preconditioning term $\mathbf{G}_t\mathbf{G}_t^\top$ captures the noise structure. Unlike conventional noise prediction, which assumes isotropic noise, our formulation enables stable training under

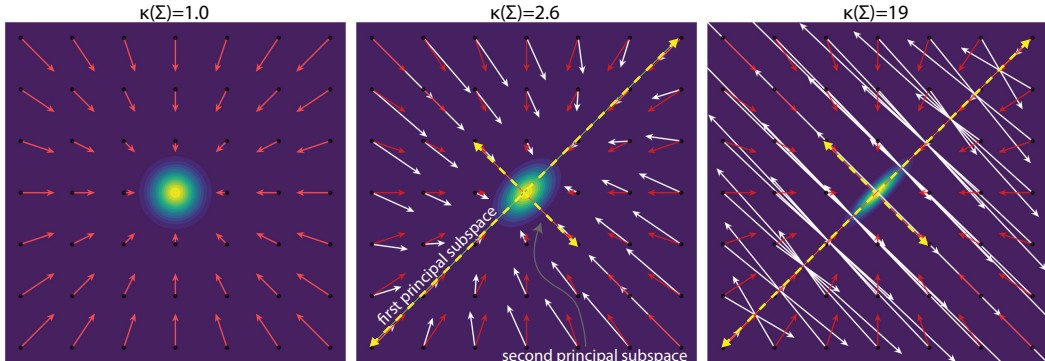

Figure 3: $\nabla_{\mathbf{x}_t} \log p(\mathbf{x}_t \mid \mathbf{x}_0)$ vector field (white) and $\mathbf{G}_t\mathbf{G}_t^\top \nabla_{\mathbf{x}_t} \log p(\mathbf{x}_t \mid \mathbf{x}_0)$ vector field (red) for increasingly anisotropic 2D Gaussian probability transition kernel $p(\mathbf{x}_t \mid \mathbf{x}_0)$. The covariance amplifies the magnitude of the conditional score field by its condition number $\kappa(\mathbf{\Sigma})$, and additionally rotates the direction towards the first principal subspace where there is higher density, while the $\mathbf{G}_t\mathbf{G}_t^\top \nabla_{\mathbf{x}_t} \log p(\mathbf{x}_t \mid \mathbf{x}_0)$ field remains stable in magnitude and directionally isotropic pointing towards the mean $\boldsymbol{\mu}_t$ of the probability path.

arbitrary Gaussian forward processes. Furthermore, Eq. 15 reveals that the conditional FM vector field $\mathbf{u}_t$ is a linear combination of our conditional WS function and the drift term (see Appendix B), highlighting that both WS and FM avoid covariance inversion by preconditioning the score function with $\mathbf{G}_t\mathbf{G}_t^\top$. This shared property enables principled modeling of flexible Gaussian probability paths.

### 3.2   Interpretation of WS

Concurrently with [4], [8] derived identical results for the reverse-time SDE, Eq. 2a, by decomposing the diffusion term of a reverse-time SDE into a unique sum of a zero-mean martingale and a *predictable process* $\mathbf{n}_t$, given as

$$\mathbf{n}_t = \frac{\sum_{i=1}^m \frac{\partial}{\partial x_t^i} \sum_{k=1}^m G_t^{ik}(\mathbf{x}_t, t) G_t^{\cdot k}(\mathbf{x}_t, t) p_t(\mathbf{x}_t)}{p_t(\mathbf{x}_t)}. \tag{14}$$

When $\mathbf{G}_t$ is independent of the state $\mathbf{x}_t$, $\mathbf{n}_t$ simplifies to $\mathbf{G}_t\mathbf{G}_t^\top \nabla_{\mathbf{x}_t} \log p_t(\mathbf{x}_t)$. This process is conditionally deterministic with respect to the filtration of the reverse time flow, motivating modeling the complete predictable process instead of the score function in isolation.

Furthermore, multiplying the score with $\mathbf{G}_t\mathbf{G}_t^\top$ whitens its vector field, as seen in Fig. 3 leading to a two-fold effect. Firstly, the original score vector field is numerically unstable; its values are highly sensitive to small errors in the residual, characterized by the condition number $\kappa(\mathbf{\Sigma})$. This leads to unstable model training, as there is often noise amplified by the condition number. Multiplying the field with $\mathbf{G}\mathbf{G}^\top$, a scalar multiple of the transition kernel's covariance, preconditions the field.

Secondly, the score field rotates in the direction towards the major principal axis that contains most of the density for the noise *transition kernel* $p(\mathbf{x}_t \mid \mathbf{x}_0)$. For anisotropic Gaussian transition kernels, the score does not point towards the data distribution, but rather towards the major principal axis of the (correlated) *noise* from the forward-time SDE. Eq. 2 naturally re-orients the field towards the data mean, providing motivation for modeling the *complete predictable process* instead of solely the score function. Furthermore, by learning the predictable process, we enable a more general scheme for SDE-based DMs by developing a model that will always have isotropic reverse-time sample paths without needing to specify the diffusion matrix during sampling.

**Connection to FM**   To connect WS DMs with FM and explain why training models with arbitrary Gaussian probability paths is achieved, we re-frame FM with the SDE framework and rewrite the conditional FM vector field expressed in terms of the VP-SDE variables in Eq. 8:

$$\mathbf{u}_t(\mathbf{x}_t \mid \mathbf{x}_0) = \mathbf{F}_t(2\mathbf{x}_t - \alpha_t\mathbf{x}_0) + \mathbf{G}_t\mathbf{G}_t^\top \nabla_{\mathbf{x}_t} \log p(\mathbf{x}_t \mid \mathbf{x}_0). \tag{15}$$

Eq. 15 reveals that the conditional FM vector $\mathbf{u}_t$ is a linear combination of our conditional WS function and the drift term (see Appendix B for derivation). The key property shared by FM and WS

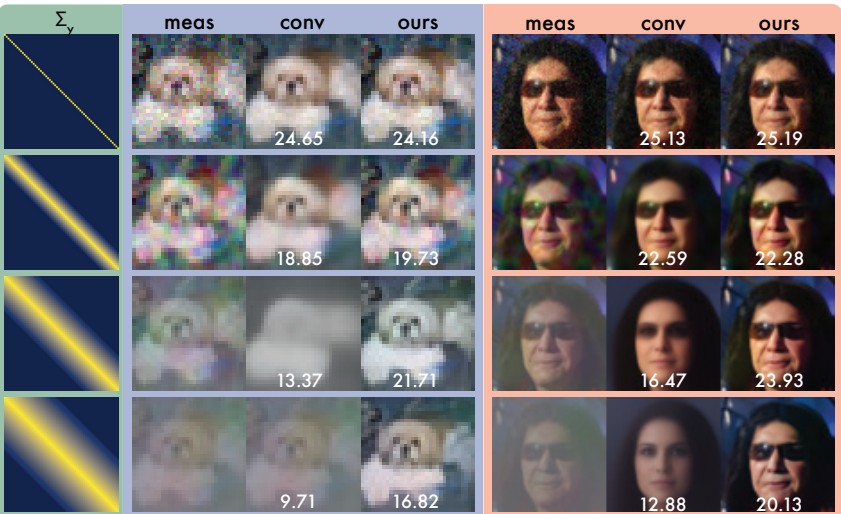

Figure 4: Measurement noise with different covariance matrices shown in the bottom left of the measurement. The PSNR is shown in white text for a sample image in the CIFAR10 validation dataset and the CelebA ($64 \times 64$) validation dataset. Compared to DMs trained only on isotropic Gaussian noise, our WS model is able to denoise correlated noise with superior PSNR. For uncorrelated noise (first row), our model has similar performance as conventional DMs.

DMs is that they avoid inverting the covariance matrix in the score by preconditioning it with $\mathbf{G}_t \mathbf{G}_t^\top$, enabling flexible modeling of arbitrary Gaussian probability paths.

## 3.3 WS diffusion priors for imaging inverse problems

We solve the imaging inverse problem using Eq. 7 with our WS diffusion prior and an approximation of the measurement likelihood score. Recall from Section 2.4 the myriad of methods developed to approximate the measurement likelihood score, all of which follow the template,

$$\nabla_{\mathbf{x}_t} \log p(\mathbf{y} \mid \mathbf{x}_t) \approx \boldsymbol{\Sigma}_{\mathbf{y}}^{-1} \nabla_{\mathbf{x}_t} \mathbf{r}(\mathbf{x}_t), \quad (16)$$

where $\nabla_{\mathbf{x}_t} \mathbf{r}(\mathbf{x}_t)$ is the gradient of the residual function that guides the update $\mathbf{x}_t$ towards regions where the observation $\mathbf{y}$ is more likely.

---

**Algorithm 1** WS diffusion priors for imaging inverse problems

---

**Require:** $T$, $\mathbf{A}$, $\mathbf{y}$, $\{\beta_t\}_{t=0}^T$, $\mathbf{n}_{\boldsymbol{\theta}}$
1: Initialize $\mathbf{x}_T \sim \mathcal{N}(\mathbf{0}, \mathbf{I})$
2: **for** $t = T$ to $0$ **do**
3:      $\mathbf{x}_t' \leftarrow (2 - \sqrt{1 - \beta_t \Delta t})\mathbf{x}_t + \frac{\mathbf{n}_{\boldsymbol{\theta}}(\mathbf{x}_t, t)\Delta t}{2}$
4:      $\mathbf{x}_{t-1} \leftarrow \mathbf{x}_t' - \lambda_t \frac{\beta_t \mathbf{A}^H (\mathbf{y} - \mathbf{A}\mathbf{x}_t)}{2}$
5: **end for**
6: **return** $\mathbf{x}_0$

---

The reverse-time SDE framework aids inverse problems with correlated noise as the diffusion matrix $\mathbf{G}_t \mathbf{G}_t^\top$ preconditions the inverse measurement covariance in the likelihood score, when $\mathbf{G}_t \mathbf{G}_t^\top$ is designed to be proportional to the covariance matrix,

$$\mathbf{G}_t \mathbf{G}_t^\top \nabla_{\mathbf{x}_t} \log p(\mathbf{y}|\mathbf{x}_t) \approx \underbrace{\mathbf{G}_t \mathbf{G}_t^\top \boldsymbol{\Sigma}_{\mathbf{y}}^{-1}}_{\propto \mathbf{I}} \nabla_{\mathbf{x}_t} \mathbf{r}(\mathbf{x}_t). \quad (17)$$

In designing our diffusion process, we set $\mathbf{G}_t \mathbf{G}_t^\top = \beta_t \mathbf{K}\mathbf{K}^\top + \gamma^2 \mathbf{I}$, where $\mathbf{K}\mathbf{K}^\top$ encompasses a large set of measurement noise covariances, and $\gamma^2$ is drawn uniformly between 0 and 1 in order to encourage the model to learn finer detailed features that reside in high spatial frequency subspaces.

In practice, a regularization term $\lambda_t$ is important to balance the generative prior with the data likelihood. For proof-of-concept, we experiment with the likelihood-guided sampling via posterior score approximation in [22] due to its functional simplicity $\nabla_{\mathbf{x}_t} \log p(\mathbf{y} \mid \mathbf{x}_t) \approx \boldsymbol{\Sigma}_{\mathbf{y}}^{-1} \mathbf{A}^H (\mathbf{y} - \mathbf{A}\mathbf{x}_t)$. We also use the deterministic sampler of Eq. 7b. The resulting algorithm is shown in Algorithm 1.

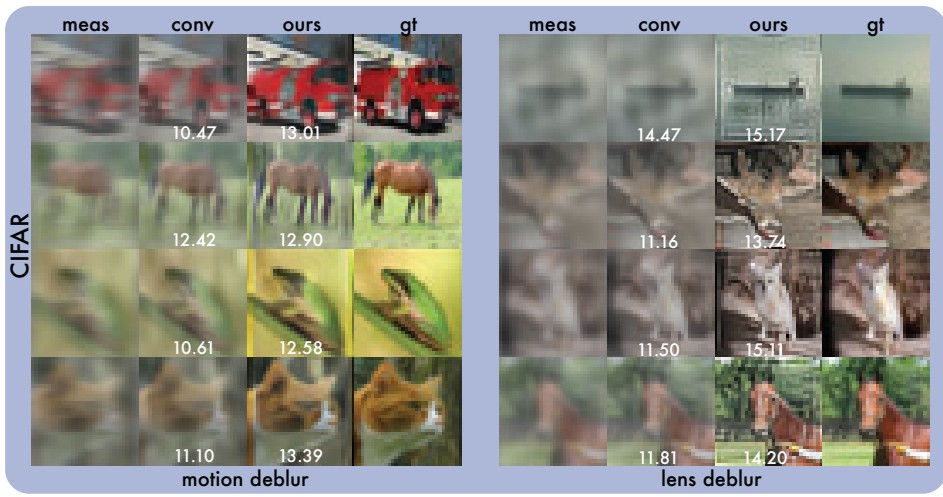

Figure 5: Motion and lens deblurring on CIFAR10 dataset with additive spatially correlated grayscale Gaussian noise of $\text{std} = 2.5$ pixels. WS diffusion prior consistently removes correlated noise resulting in higher PSNR compared to DMs trained solely on isotropic Gaussian noise.

## 4 Experiments

### 4.1 Training details

For each dataset, we train two attention UNet models based on the architecture in [18] with three residual blocks in each downsampling layer, where one is for the conventional isotropic Gaussian SDE, and the other our anisotropic Gaussian SDE. We set the learning rate to $3\text{e}^{-5}$ with a linear decay schedule. For CIFAR10 ($32 \times 32$), the batch size is 128, for CelebA ($64 \times 64$), the batch size is 16, and for CelebA-HQ ($256 \times 256$), the batch size is 4. Models were trained on a single NVIDIA L40S GPU with 48GB of memory for two days. Our model is trained on the training sets of CIFAR-10 [29], CelebA ($64 \times 64$) [36] and CelebA-HQ ($256 \times 256$) [24] where $\mathbf{K}$ is a 2D Gaussian convolutional matrix characterized by an $\text{std}$. For CIFAR, the $\text{std}$ that characterizes $\mathbf{K}$ is uniformly distributed between 0.1 and 3, between 0.1 and 5 for CelebA ($64 \times 64$), and between 0.1 and 20 for CelebA-HQ ($256 \times 256$) where $\text{std} \leq 0.5$ equals the 2D delta function. The noise is also randomly grayscale or color with a 0.5 probability.

### 4.2 Imaging inverse problems with correlated noise

It is well-established that natural image spectra exhibit exponential decay [57], indicating the dominance of low-frequency components in representing images. When additive measurement noise occupies the same spectral subspace, especially at low frequencies, the computational imaging task becomes fundamentally more challenging. We show that our framework is beneficial as a generative prior for solving inverse problems with such structured noise by experimenting with a variety of computational imaging modalities that are known to be affected by structured noise.

The measurements in our experiments are corrupted by additive grayscale structured noise, designed to mimic real-world conditions frequently encountered in both computational photography—such as fog, haze, and atmospheric turbulence—and computational microscopy—including fluorescence background, laser speckle, and detector noise. We use Algorithm 1 with $T = 1000$ and $\beta_{min} = 0.01$ and $\beta_{max} = 20$ so that the SNR decays to 0 at $t = T$. For results with our WS prior, $\mathbf{x}_T$ was drawn from $\mathcal{N}(\mathbf{0}, \mathbf{KK}^\top)$ with $\text{std} = 3$ and $\text{std} = 6$ for CIFAR and CelebA, respectively with grayscale color (all color channels have the same value). For conventional DM prior results $\mathbf{x}_T$ was drawn from $\mathcal{N}(\mathbf{0}, \mathbf{I})$. All evaluation was performed on unseen validation dataset sample images picked uniformly at random. The regularization parameter $\lambda$ scales the magnitude of the likelihood step to be proportional to the magnitude of the prior step as was done by [22]. Line search was used to find an optimal $\lambda$ that yielded a reconstruction with the highest PSNR, where PSNR is defined as

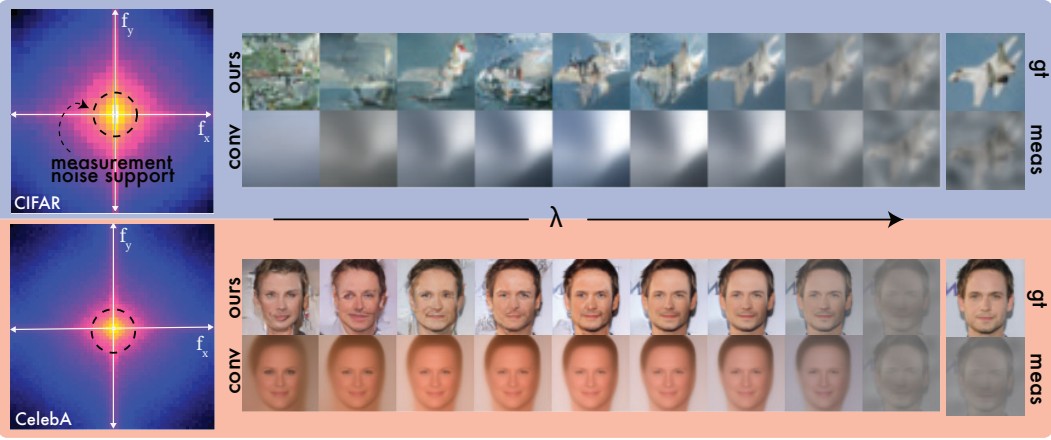

Figure 6: Effect of changing regularization parameter $\lambda$ for denoising. The top figures are the average power spectral density of the images in CIFAR and CelebA, with a dotted red circle to denote the frequency support of the additive correlated noise. Changing the regularization weight $\lambda$ for denoising affects the final reconstructions using our WS diffusion priors (top) and conventional diffusion priors (bottom). When $\lambda$ is 0, it is equivalent to sampling form $p(\mathbf{x})$. As $\lambda$ increases, the generative modeling effect is overpowered by the measurement fidelity term, that the reconstruction resembles the measurement $\mathbf{y}$.

$\mathrm{PSNR} = 20 \cdot \log_{10}\left(\frac{1}{\mathrm{MSE}}\right)$ where $\mathrm{MSE}$ is the mean squared error between the reconstruction and the ground truth.

Our results are demonstrated on a variety of computational imaging tasks such as imaging through fog, motion deblurring, lens deblurring, linear inverse scattering, and differential defocus. More details are in Appendix C.

**Denoising correlated noise**   To demonstrate the capabilities of our model as a generative prior for measurements corrupted by correlated noise, we explore the denoising problem and compare the results with that of a conventional score-based diffusion prior that was trained only on isotropic Gaussian forward diffusion. Fig. 2 shows the results on CIFAR and CelebA ($64 \times 64$) test samples across a range of SNRs, where color is faithfully restored from fog-like corruption and likeness to the dataset is maintained due to the generative prior.

**Generalize to different noise structures**   Our model generalizes to different measurement noise covariance matrices with varying Gaussian noise distributions. Fig. 4 reveals that measurements corrupted by different distributions of spatially correlated Gaussian noise are restored with higher PSNR compared to conventional DM priors (conv). Conventional score-based priors change the higher level semantic features of the measurement, due to the model's inability to distinguish noisy features from target image features based on Fourier support. Specifically, the added correlated noise's low frequency support overlaps with that of the visual features in the data, seen in Fig. 6. This makes the reconstruction task more difficult.

**Spectral inductive bias**   WS diffusion priors more effectively distinguish structured noise from target features compared to conventional diffusion priors trained on isotropic Gaussian noise. As shown in Fig. 6, standard DMs tend to suppress high-frequency components in the measurement, assuming they originate from noise—a valid assumption only when the noise Fourier support extends beyond that of the data. For CIFAR, whose average signal spectrum extends beyond the noise's, this misclassification leads to undesired attenuation of image features. In contrast, for CelebA ($64 \times 64$), where the average image spectrum lies within the noise support, conventional models better preserve image features.

WS DMs, trained on ensembles of Gaussian trajectories, learn to identify structured noise beyond simple spectral heuristics. This enables selective removal of low-frequency noise even when it spectrally overlaps with signal content, yielding improved denoising performance in the presence of correlated noise.

**Computational imaging**    Using WS diffusion prior to solve inverse problems with non-identity forward operators outperforms traditional score-based diffusion priors in PSNR. Noticeably, our diffusion prior is able to maintain fidelity to the color distribution for restoring measurements corrupted by grayscale fog-like noise, while conventional score-based diffusion priors fail to remove the noise, as seen in Figs. 5 and 8 for deblurring inverse problems. Additional results for other imaging inverse problems on CIFAR, as well as on the CelebA ($64 \times 64$) and CelebA-HQ ($256 \times 256$) datasets, are presented in Appendix C and and Figs. 7, 8, 9, 10, and 11.

## 5    Conclusion

We introduced WS diffusion, a generalization of score-based methods that learns the Whitened Score, $\mathbf{G}_t \mathbf{G}_t^\top \nabla_{\mathbf{x}_t} \log p_t(\mathbf{x}_t)$. This avoids noise covariance inversion, enabling arbitrary anisotropic Gaussian forward processes and bridging connections to FM. We demonstrate WS diffusion as robust generative priors for inverse problems involving correlated noise, common in computational imaging. Experiments consistently showed superior PSNR and visual reconstructions compared to conventional diffusion priors trained on isotropic noise, particularly in accurately handling structured noise while preserving image features. WS diffusion provides a principled approach for developing effective generative models tailored to structured noise, advancing their utility in computational imaging applications.

**Limitations and future work**    A primary limitation of our approach lies in the computational cost of sampling. The current time discretization of the reverse-time SDE necessitates approximately 1000 denoising steps, which may be prohibitive for certain practical applications. Reducing the number of denoising steps through model distillation represents a promising direction for future work [44, 51].

Another limitation concerns the absence of an explicit mechanism to estimate the measurement noise covariance, which directly influences the specification of the diffusion matrix $\mathbf{G}_t$. A natural extension of this framework would involve parameterizing and learning $\mathbf{G}_t$ jointly with the model parameters. Such an approach would allow the diffusion process to adaptively capture data-dependent or task-specific noise structures, thereby enhancing the model's flexibility and representational capacity. This line of work connects to recent advances in vector-valued and multivariate diffusion models, which have demonstrated improved performance in scenarios characterized by complex or structured noise.

Finally, while our model exhibits strong performance as a denoising prior, additional research is needed for WS DMs to achieve competitive results in unconditional or conditional generation tasks. Promising directions include latent diffusion formulations and related techniques for improving generative efficiency and expressiveness [12, 42, 41].

**Acknowledgements** We are grateful for a grant from 5022 - Chan Zuckerberg Initiative DAF, an advised fund of Silicon Valley Community Foundation. We also thank the Boston University Shared Computing Cluster for computational resources. J.A. acknowledges funding from the NSF Graduate Research Fellowship Program (GRFP) under Grant No. 2234657.

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

# A Denoising Whitened Score matching

**Lemma A.1.** *(Generalized Tweedie's formula for non-diagonal covariance). Let:*

$$\mathbf{x}_t = \alpha_t \mathbf{x}_0 + \mathbf{z}, \tag{18}$$

*where $\mathbf{x}_0 \sim p(\mathbf{x})$ and $\mathbf{z} \sim \mathcal{N}(\mathbf{0}, \boldsymbol{\Sigma})$. Then*

$$\alpha_t \mathbb{E}[\mathbf{x}_0 \mid \mathbf{x}_t] = \mathbf{x}_t + \boldsymbol{\Sigma} \nabla_{\mathbf{x}_t} \log p_t(\mathbf{x}_t) \tag{19}$$

*Proof.*

$$
\begin{aligned}
\nabla_{\mathbf{x}_t} \log p_t(\mathbf{x}_t) &= \frac{\nabla_{\mathbf{x}_t} p_t(\mathbf{x}_t)}{p_t(\mathbf{x}_t)} \\
&= \frac{1}{p_t(\mathbf{x}_t)} \nabla_{\mathbf{x}_t} \int p_t(\mathbf{x}_t, \mathbf{x}_0)\, \mathrm{d}\mathbf{x}_0 && \text{(de-marginalize joint distribution)} \\
&= \frac{1}{p_t(\mathbf{x}_t)} \nabla_{\mathbf{x}_t} \int p_t(\mathbf{x}_t \mid \mathbf{x}_0) p_0(\mathbf{x}_0)\, \mathrm{d}\mathbf{x}_0 && \text{(factor joint via Bayes rule)} \\
&= \frac{1}{p_t(\mathbf{x}_t)} \int \nabla_{\mathbf{x}_t} p_t(\mathbf{x}_t \mid \mathbf{x}_0) p_0(\mathbf{x}_0)\, \mathrm{d}\mathbf{x}_0 && \text{(move gradient inside integral)} \\
&= \frac{1}{p_t(\mathbf{x}_t)} \int p_t(\mathbf{x}_t \mid \mathbf{x}_0) \nabla_{\mathbf{x}_t} \log p_t(\mathbf{x}_t \mid \mathbf{x}_0) p_0(\mathbf{x}_0)\, \mathrm{d}\mathbf{x}_0 && \text{(use identity } \nabla f = f \nabla \log f) \\
&= \int p_0(\mathbf{x}_0 \mid \mathbf{x}_t) \boldsymbol{\Sigma}^{-1} (\alpha_t \mathbf{x}_0 - \mathbf{x}_t)\, \mathrm{d}\mathbf{x}_0 && \text{(Gaussian conditional score)} \\
&= \boldsymbol{\Sigma}^{-1} (\alpha_t \, \mathbb{E}[\mathbf{x}_0 \mid \mathbf{x}_t] - \mathbf{x}_t) && \text{(expectation under posterior)}
\end{aligned}
$$

$\square$

**Theorem A.2.** *(Denoising Whitened Score matching)*
*Our Whitened Score matching loss function, Eq. 13, copied here as:*

$$\mathbb{E}_{t \sim U(0,1], \mathbf{x}_t \sim p(\mathbf{x}_t \mid \mathbf{x}_0), \mathbf{x}_0 \sim p(\mathbf{x})} \left\{ \|\mathbf{n}_\theta(\mathbf{x}_t, t) - \mathbf{G}_t \mathbf{G}_t^\top \nabla_{\mathbf{x}_t} \log p_t(\mathbf{x}_t \mid \mathbf{x}_0)\|_2^2 \right\} \tag{20}$$

*is a denoising objective that uses the conditional probability to estimate $\mathbf{G}_t \mathbf{G}_t^\top \nabla_{\mathbf{x}_t} \log p_t(\mathbf{x}_t)$. Here we prove that our loss function results in an estimator for $\mathbf{G}_t \mathbf{G}_t^\top \nabla_{\mathbf{x}_t} \log p_t(\mathbf{x}_t)$.*

*Proof.* Let $p(\mathbf{x}_t \mid \mathbf{x}_0)$ denote the Gaussian probability transition kernel associated with the forward-time SDE in Eq. 8. For a linear SDE in $\mathbf{x}_t$, the covariance $\boldsymbol{\Sigma}_t$ of the transition kernel is a scalar multiple of twice the diffusion matrix, $\mathbf{G}_t \mathbf{G}_t^\top$ ([54]) provided the initial conditions of $\boldsymbol{\mu}(0) = \mathbf{x}_0$ and $\boldsymbol{\Sigma}(0) = \mathbf{0}$ for $p(\mathbf{x}_t \mid \mathbf{x}_0)$:

$$\boldsymbol{\Sigma}_t = c \mathbf{G}_t \mathbf{G}_t^\top. \tag{21}$$

The Minimum Mean Squared Estimator (MMSE) $\mathbb{E}[\mathbf{x}_0 \mid \mathbf{x}_t]$ is achieved through optimizing the least squares objective:

$$\min_{\boldsymbol{\theta}} \mathbb{E}_{\mathbf{x}_t \sim p_t(\mathbf{x}_t \mid \mathbf{x}_0), \mathbf{x}_0 \sim p(\mathbf{x})} \left[ \|\mathbf{h}_{\boldsymbol{\theta}}(\mathbf{x}_t) - \mathbf{x}_0\|_2^2 \right], \tag{22}$$

such that $\mathbf{h}_{\boldsymbol{\theta}^*}(\mathbf{x}_t) = \mathbb{E}[\mathbf{x}_0 \mid \mathbf{x}_t]$, for optimal network parameters $\boldsymbol{\theta}^*$. Tweedie's formula from Lemma A.1 gives us that,

$$\alpha_t \mathbf{h}_{\boldsymbol{\theta}^*}(\mathbf{x}_t) = \mathbf{x}_t + \boldsymbol{\Sigma} \nabla_{\mathbf{x}_t} \log p_t(\mathbf{x}_t). \tag{23}$$

Parameterizing $\mathbf{h}_{\boldsymbol{\theta}}(\mathbf{x}_t)$ as

$$\mathbf{h}_{\boldsymbol{\theta}}(\mathbf{x}_t) = \frac{\mathbf{x}_t + c \mathbf{n}_{\boldsymbol{\theta}}(\mathbf{x}_t, t)}{\alpha_t}, \tag{24}$$

with scalars $c, \alpha_t \in \mathbb{R}$ and $\mathbf{n}_{\boldsymbol{\theta}}(\mathbf{x}_t, t)$ our $\mathbf{G}_t \mathbf{G}_t^\top \nabla_{\mathbf{x}_t} \log p_t(\mathbf{x}_t)$ model, the MMSE objective in Eq. 22 becomes

$$\min_{\boldsymbol{\theta}} \mathbb{E}_{\mathbf{x}_t \sim p_t(\mathbf{x}_t \mid \mathbf{x}_0), \mathbf{x}_0 \sim p(\mathbf{x})} \left[ \|\mathbf{x}_t + c \mathbf{n}_{\boldsymbol{\theta}}(\mathbf{x}_t, t) - \alpha_t \mathbf{x}_0\|_2^2 \right], \tag{25}$$

which is equivalent to our objective in Eq. 20 through the closed form expression of $\mathbf{G}_t \mathbf{G}_t^\top \nabla_{\mathbf{x}_t} \log p_t(\mathbf{x}_t \mid \mathbf{x}_0)$ given in Eq. 12. Finding the optimal $\boldsymbol{\theta}^*$ implies

$$\mathbf{h}_{\boldsymbol{\theta}^*}(\mathbf{x}_t) = \mathbb{E}[\mathbf{x}_0 \mid \mathbf{x}_t] = \frac{\mathbf{x}_t + c\mathbf{n}_{\boldsymbol{\theta}^*}(\mathbf{x}_t, t)}{\alpha_t} = \frac{\mathbf{x}_t + \boldsymbol{\Sigma}\nabla_{\mathbf{x}_t}\log p_t(\mathbf{x}_t)}{\alpha_t}. \tag{26}$$

which proves that our model $\mathbf{n}_{\boldsymbol{\theta}}(\mathbf{x}_t, t)$ learns $\mathbf{G}_t\mathbf{G}_t^\top\nabla_{\mathbf{x}_t}\log p_t(\mathbf{x}_t)$ with objective Eq. 20.

$\square$

## B  Flow matching in SDE

Consider the probability path
$$p_t = \mathcal{N}(\mathbf{x}_t \mid \boldsymbol{\mu}_t(\mathbf{x}_0), \boldsymbol{\Sigma}_t(\mathbf{x}_0)), \tag{27}$$
and the corresponding continuous normalizing flow:
$$\phi_t(\mathbf{x}_0) = \boldsymbol{\mu}_t(\mathbf{x}_0) + \boldsymbol{\Sigma}_t^{\frac{1}{2}}(\mathbf{x}_0), \tag{28}$$
where we define $\boldsymbol{\Sigma}_t^{\frac{1}{2}}(\mathbf{x}_0)$ such that $\mathrm{Cov}[\phi_t(\mathbf{x}_0)] = \boldsymbol{\Sigma}_t(\mathbf{x}_0) = \boldsymbol{\Sigma}_t^{\frac{1}{2}}(\mathbf{x}_0)(\boldsymbol{\Sigma}_t^{\frac{1}{2}}(\mathbf{x}_0))^\top$ and the initial condition $\boldsymbol{\mu}_0(\mathbf{x}_0) = \mathbf{x}_0$. This probability path is equivalent to the probability transition kernel in Eq. 9 defined by a linear SDE Eq. 8 with drift coefficient $\mathbf{F}_t$ and diffusion matrix $\mathbf{G}_t$. Therefore we may attain the time derivatives of the mean and covariance functions using Fokker-Planck (see Eqs. 6.2 in [54]) expressed as

$$\boldsymbol{\Sigma}_t'(\mathbf{x}_0) = 2\mathbf{F}_t\boldsymbol{\Sigma}_t(\mathbf{x}_0) + \mathbf{G}_t\mathbf{G}_t^\top, \tag{29}$$

$$\boldsymbol{\mu}_t'(\mathbf{x}_0) = \mathbf{F}_t\boldsymbol{\mu}_t(\mathbf{x}_0). \tag{30}$$

The FM conditional vector field for Gaussian probability paths is

$$\mathbf{u}_t(\mathbf{x}_t \mid \mathbf{x}_0) = \boldsymbol{\Sigma}_t'(\mathbf{x}_0)\boldsymbol{\Sigma}_t^{-1}(\mathbf{x}_0)(\mathbf{x}_t - \boldsymbol{\mu}_t(\mathbf{x}_0)) + \boldsymbol{\mu}_t'(\mathbf{x}_0). \tag{31}$$

Plugging in Eqs. 30 and 29 into Eq. 31 we have

$$\mathbf{u}_t(\mathbf{x}_t \mid \mathbf{x}_0) = \boldsymbol{\Sigma}_t'(\mathbf{x}_0)\boldsymbol{\Sigma}_t^{-1}(\mathbf{x}_0)(\mathbf{x}_t - \boldsymbol{\mu}_t(\mathbf{x}_0)) + \boldsymbol{\mu}_t'(\mathbf{x}_0) \tag{32}$$

$$= (2\mathbf{F}_t + \mathbf{G}_t\mathbf{G}_t^\top\boldsymbol{\Sigma}_t^{-1}(\mathbf{x}_0)(\mathbf{x}_t - \alpha_t\mathbf{x}_0)) + \mathbf{F}_t\boldsymbol{\mu}_t(\mathbf{x}_0) \tag{33}$$

$$= 2\mathbf{F}_t\mathbf{x}_t - \mathbf{F}_t\alpha_t\mathbf{x}_0 + \mathbf{G}_t\mathbf{G}_t^\top\boldsymbol{\Sigma}_t^{-1}(\mathbf{x}_0)(\mathbf{x}_t - \alpha_t\mathbf{x}_0) \tag{34}$$

$$= \mathbf{F}_t(2\mathbf{x}_t - \alpha_t\mathbf{x}_0) - \mathbf{G}_t\mathbf{G}_t^\top\nabla_{\mathbf{x}_t}\log p(\mathbf{x}_t \mid \mathbf{x}_0). \tag{35}$$

## C  Imaging inverse problems

**Imaging through fog/turbulence**  We simplify imaging through fog/turbulence as a denoising problem for correlated noise which we achieve by setting the imaging system $\mathbf{A} = \mathbf{I}$. Specifically, we demonstrate our model for grayscale low-pass filtered white Gaussian noise, where $\mathbf{K}$ is a 2D Gaussian kernel characterized by a std.

In our method, the model is trained to denoise correlated noise, and is able to distinguish target image features from anisotropic Gaussian noise features, even though both may share similar spatial frequency support. In contrast, the conventional DM trained to denoise only isotropic Gaussian noise, removes the images features for low enough $\lambda$, as it mistakes the correlated additive noise for the target features. As seen in Fig. 6, there are more image features on average outside the additive noise support for CIFAR, and for CelebA, the support of the image features are more closely overlapped with the noise support. Additional result on the CelebA-HQ ($256 \times 256$) dataset is shown in Fig. 7 .

**Motion deblurring**  Motion blur is a common image degradation in computational photography. We experiment with a spatially invariant horizontal motion blur kernel of five pixels for CIFAR and seven for CelebA. The additive correlated noise is Gaussian-filtered grayscale WGN with a circular kernel of $\mathrm{std} = 2.5$ pixels for CIFAR and 5 pixels for CelebA, each with $\mathrm{SNR} = 0.493$. The result is shown in Figs. 5 and 8.

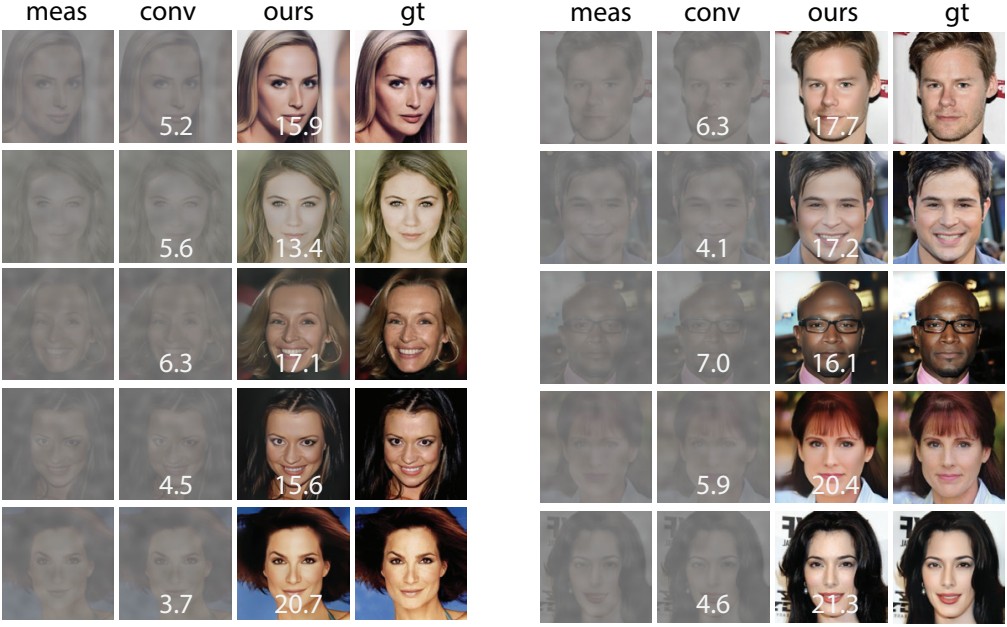

Figure 7: Denoising correlated noise on CelebA-HQ ($256 \times 256$). We benchmark our WS DM trained on anisotropic Gaussian noise with the conventional DM (conv) trained on isotropic Gaussian noise. Results for measurements $\mathbf{y}$ with additive grayscale Gaussian noise of $\mathrm{std} = 5$ pixels.

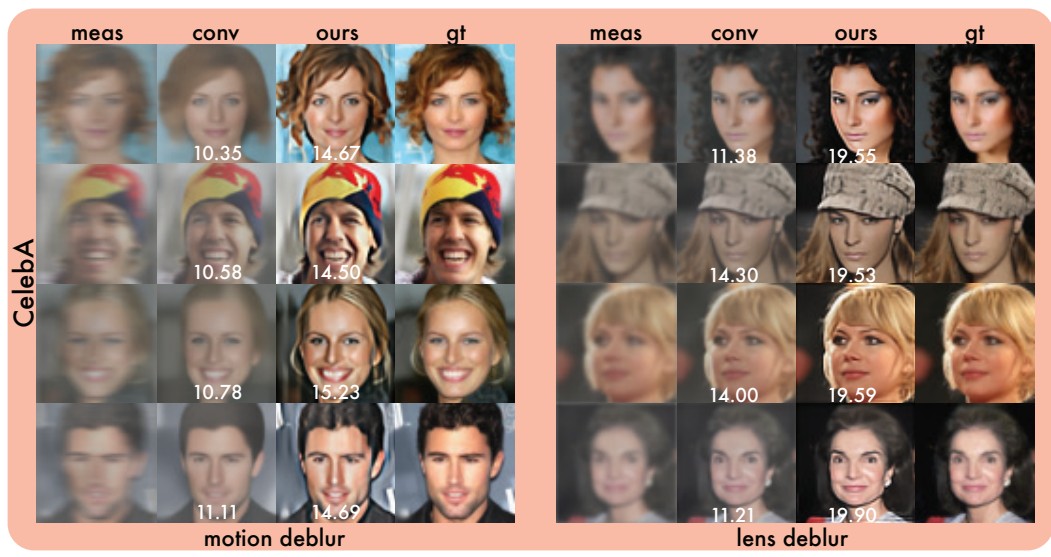

Figure 8: Motion and lens deblurring on CIFAR10 dataset with additive spatially correlated grayscale Gaussian noise of $\mathrm{std} = 2.5$ pixels. Our diffusion prior is able to consistently remove correlated noise resulting in superior PSNR compared to DMs trained solely on isotropic Gaussian noise.

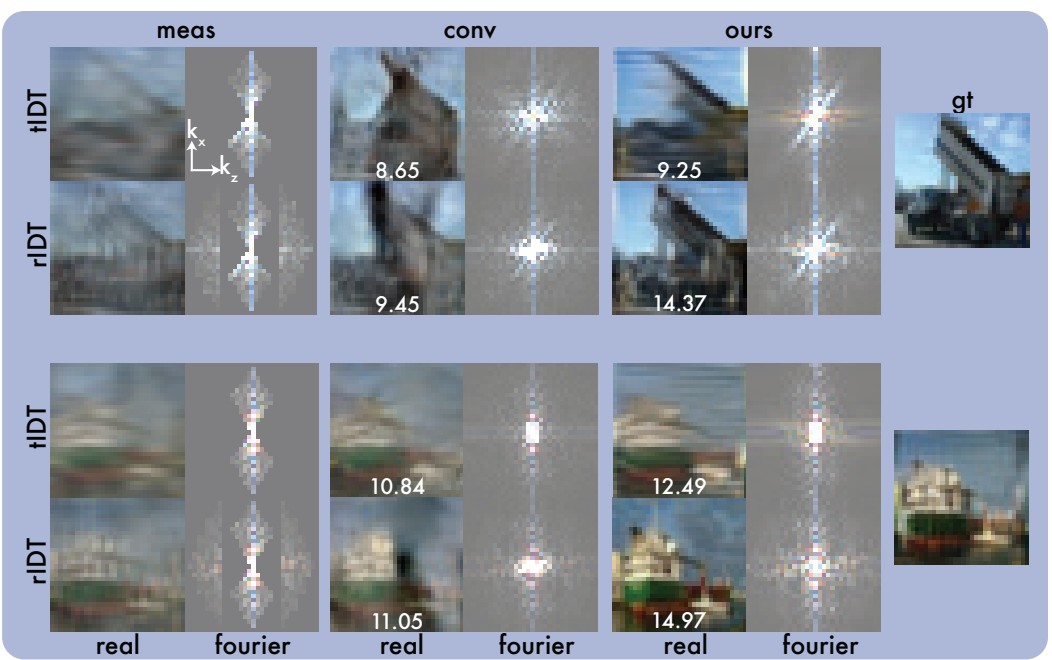

Figure 9: Linear inverse scattering CIFAR

**Lens deblurring** Lens blur is the loss of high spatial frequency information as a result of light rays being focused imperfectly due to the finite aperture size, causing rays from a point source to spread over a region in the image plane rather than converging to a single point. This can be effectively modeled as a convolution between a circular Gaussian kernel and the clean image. In Figs. 5 and 8, we demonstrate our WS diffusion prior on lens deblurring with a Gaussian blur kernel of $STD = 0.8$ and $1.0$ for CIFAR and CelebA, respectively. The additive correlated noise is Gaussian-filtered grayscale WGN with a circular kernel of $std = 2.5$ pixels for CIFAR and 5 pixels for CelebA, each with $SNR = 0.810$.

**Linear inverse scattering** Inverse scattering is a prevalent direction in optical imaging, to recover the permittivity field from measurements under angled illumination. Intensity diffraction tomography (IDT) is a powerful computational microscopy technique that can recover 3D refractive index distribution given a set of 2D measurements. The model can be linearized using the first Born approximation ([32]):

$$\mathbf{u}(\mathbf{r}) = \mathbf{u}_i(\mathbf{r}) + \int \mathbf{u}_i(\mathbf{r}')\mathbf{V}(\mathbf{r}')\mathbf{G}(\mathbf{r} - \mathbf{r}')d\mathbf{r}', \tag{36}$$

for the field at the measurement plane $\mathbf{u}(\mathbf{r})$ and incident field $\mathbf{u}_i(\mathbf{r})$. The scattering potential $\mathbf{V}(\mathbf{r}) = \frac{1}{4\pi}k_0^2\Delta\epsilon(\mathbf{r})$ with permittivity contrast $\Delta\epsilon(\mathbf{r}) = \epsilon(\mathbf{r}) - \epsilon_0$ between the sample $\epsilon(\mathbf{r})$ and surrounding medium $\epsilon$, and wavenumber $k_0 = \frac{2\pi}{\lambda}$ for illumination wavelength $\lambda$. Green's function $\mathbf{G}(\mathbf{r}) = \frac{\exp(ik|\mathbf{r}|)}{\mathbf{r}}$ where $k = \sqrt{\epsilon_0}k_0$.

When the illumination is transmissive, referred to as transmission intensity diffraction tomography (tIDT), meaning that the light passes through the sample, the linear operator, $\mathbf{A}$, results in a mask in the shape of a cross section of a torus that attenuates Fourier coefficients as seen in Figs. 9 and 10, leading to the well-known "missing cone" problem.

In reflection IDT (rIDT), placing the sample object on a specular mirror substrate causes light to reflect towards the camera, enabling the capture of additional axial frequency components and partially filling the missing cone, shown in Figs. 9 and 10.

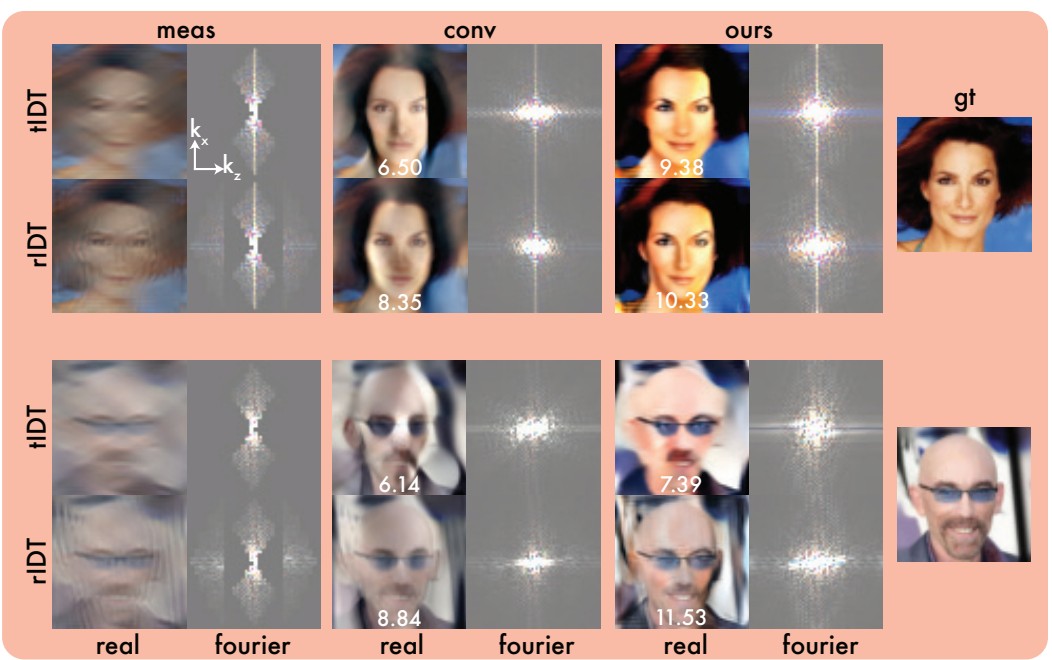

Figure 10: Linear inverse scattering CelebA ($64 \times 64$)

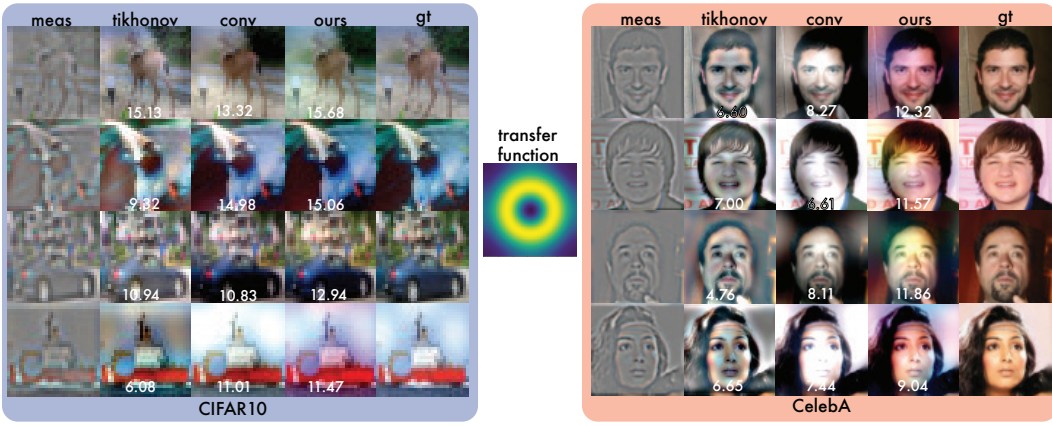

Figure 11: Laplace imaging, Transport of Intensity (TIE)

We experiment with both tIDT and rIDT with the measurements corrupted by low-pass filtered grayscale WGN to mimic background noise common in microscopy. The grayscale noise is similarly Gaussian-filtered with $\mathrm{std} = 2.5$ and 5, for CIFAR and CelebA, respectively with $\mathrm{SNR} = 0.632$.

**Differential defocus**   Differential defocus is a computational imaging technique that aims to recover the depth map from a series of defocused measurements ([2]). The linear operator can be realized with a 2D Laplacian kernel, bandpassing mid-frequency components. In computational microscopy, this is also known as transport of intensity imaging ([59]) to recover the phase and amplitude of an object.

We demonstrate our framework on the differential defocus problem in Fig. 11. The noise is again grayscale WGN filtered with Gaussian kernels of $\mathrm{std} = 2.5$ and 5, for CIFAR and CelebA, respectively with $\mathrm{SNR} = 12.91$. We also compare with a Tikhonov regularization, which is an $L_2$ norm prior on the object to constrain the energy of the reconstruction.

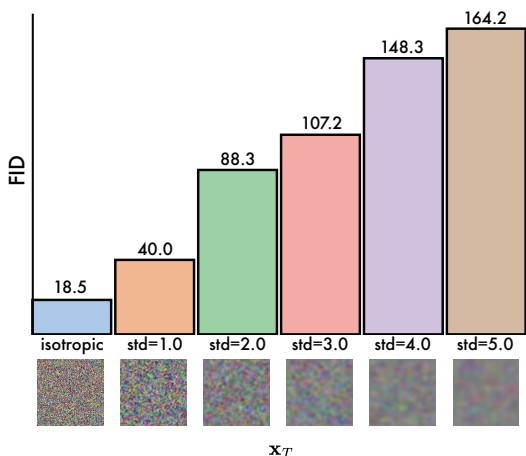

Figure 12: FID scores for different WS DMs trained on different $\text{std}_{max}$.

# D  Generative modeling

We also train 6 different models where we vary the $\text{std}$ of the maximum value of the Gaussian blur kernel, $\mathbf{K}$, from a delta function (isotropic Gaussian noise) to $\text{std}_{max} = 5$ pixels. During training, $\mathbf{K}$ varies uniformly from $\text{std} = 0.1$ to $\text{std} = \text{std}_{max}$.

Novel samples are produced by solving the probability flow ODE, replacing $\mathbf{G}_t \mathbf{G}_t^\top \nabla_{\mathbf{x}_t} \log p_t(\mathbf{x}_t)$ with our optimized model $\mathbf{n}_\theta(\mathbf{x}_t, t)$ using Euler-Maruyama discretization with $T = 1000$ and $\beta_{min} = 0.01$ and $\beta_{max} = 20$. The initial noise condition, $\mathbf{x}_T \sim \mathcal{N}(\mathbf{0}, \mathbf{K}_{\text{std}_{max}} \mathbf{K}_{\text{std}_{max}}^\top)$. The Fréchet Inception Distance (FID) scores decrease as the spatial correlation range increases as seen in Fig. 12.

While WS DMs perform well as generative denoising priors for inverse problems, we leave to future work further investigation on their generative capabilities.

# E  Forward Consistency Loss

The model $\mathbf{n}_\theta(\mathbf{x}_t, t)$ is trained to approximate the scaled noise component introduced in the forward stochastic differential equation (SDE) that perturbs $\mathbf{x}_0$ to yield $\mathbf{x}_t$. Accordingly, the score function $\mathbf{G}_t \mathbf{G}_t^\top \nabla_{\mathbf{x}_t} \log p(\mathbf{x}_t \mid \mathbf{x}_0)$ in Eq. 12 can be substituted with the model prediction. To enforce consistency with the forward diffusion process, we introduce an auxiliary loss term defined as:

$$L_2 = \mathbb{E}_{t \sim \mathcal{U}(0,1], \mathbf{x}_0 \sim p(\mathbf{x}), \mathbf{x}_t \sim p(\mathbf{x}_t | \mathbf{x}_0)} \left\{ \left\| \mathbf{x}_0 - \frac{\beta_t \mathbf{x}_t + (1 - \alpha_t^2) \mathbf{n}_\theta(\mathbf{x}_t, t)}{\beta_t \alpha_t} \right\|_2^2 \right\}. \tag{37}$$

The term inside the expectation represents a reconstruction of $\mathbf{x}_0$ based on the noisy sample $\mathbf{x}t$ and the model prediction $\mathbf{n}_\theta$. Minimizing $L_2$ encourages the model to remain faithful to the generative process defined by the forward SDE.

Empirically, we find that including $L_2$ as an auxiliary objective—weighted equally with the primary loss term $L$—leads to improved training stability and faster convergence.

