# OpenReview forum: "Whitened Score Diffusion: A Structured Prior for Imaging Inverse Problems"
_NeurIPS.cc/2025/Conference — NeurIPS 2025 poster_

### Official Review · Reviewer_nvZa · 2025-06-30

**Clarity:** 4
**Significance:** 3
**Originality:** 3
**Rating:** 4
**Confidence:** 4

**Summary:**

The paper introduces Whitened Score Diffusion, an extension to score-based diffusion models that enables stable training over arbitrary (including anisotropic) Gaussian noising processes. The key innovation is to learn the product $\mathbf{G}_t\mathbf{G}t^{\top}\nabla{\mathbf{x}_t}\log p_t(\mathbf{x}_t)$, rather than just the score, which avoids problematic covariance inversions when handling structured, non-isotropic noise. The method is motivated as a structured Bayesian prior for imaging inverse problems involving correlated noise. Experiments on CIFAR10 and CelebA demonstrate that the proposed method outperforms conventional DMs—especially under correlated/structured noise—across several computational imaging tasks.

**Questions:**

1. How sensitive are the improvements to the exact choice of spatial covariance? Could the authors provide results or analysis for highly non-circulant or non-Gaussian structures in $\mathbf{K}$?
2. Could more challenging or realistic noise models be evaluated? For example, astronomy, or non-additive processes common in optics/microscopy.
3. How robust is sampling to mis-specification of $\mathbf{G}_t$ during inference? Is there a drop in performance if the test noise structure mismatches the training structure, and can WS DMs be tuned or regularized for increased generalization?
4. Could the method be extended to non-linear/inverse problems outside the linear Gaussian setting? What would be required from an algorithmic or mathematical perspective?

**Ethical Concerns:**

["NO or VERY MINOR ethics concerns only"]

**Final Justification:**

This is a well-motivated and theoretically sound paper proposing a method to train diffusion models with arbitrary Gaussian noise for inverse problems. While I appreciate the new 256x256 results, the paper would be more convincing with additional results on downstream imaging tasks. Overall, I lean positive to this paper.

**Limitations:**

Yes.

**Paper Formatting Concerns:**

No.

**Quality:**

3

**Strengths And Weaknesses:**

**Strengths**
- This work addresses a significant limitation in current score-based diffusion models by making training feasible and stable for arbitrarily structured Gaussian noise processes. This is well-motivated for imaging scenarios where noise is rarely isotropic, as detailed in the introduction and Section 2.3.
- The visualizations in Figure 3 are straightforward, which effectively illustrate how the WS transformation preconditions the score field to be magnitude-stable and isotropic, even for highly anisotropic covariances.
- Empirical results consistently show that WS DMs provide higher PSNR and improved perceptual quality over conventional DMs in various inverse imaging tasks with structured noise.
- The theoretical analysis reveals a clear connection drawn between the new "whitened score" objective and flow matching. The relationship both justifies the method and indicates its generality.


**Weaknesses**
- The empirical validation scope is somewhat narrow: Only CIFAR10 (32×32) and CelebA (64×64) are used in experiments. While these are standard, more diverse datasets or higher-resolution tasks (e.g., FFHQ 256×256 or real-world scientific/medical applications) would strengthen the generalizability claim. This is particularly relevant as the method is framed as benefiting scientific imaging.
- Baseline selection should be expanded. Only a conventional DM trained with isotropic noise is used as a baseline. Previous work[17] on handling structured/correlated noise is referenced, but not directly compared against. There should also be flow matching-based baselines or other Bayesian generative priors.
- There is no explicit ablation on the impact of the whitening transformation versus simply adjusting training objective scaling, nor an exploration of how the choice of covariance $\mathbf{K}$ (e.g., various correlations or non-circulant structures) affects results.
- Realistic noise modeling is missing. The extent to which this matches real scientific imaging noise statistics (e.g., turbulence, CCD noise) is not quantitatively validated.

Overall, I like the idea of considering arbitrarily structured Gaussian noise processes for real-world DM-based inverse solvers, but the empirical results are insufficient in the current version. I would like to increase the score if more results are provided.

---

> ### Author Rebuttal · Authors · 2025-07-31
>
> **W1**: We thank the reviewer for this insightful comment. We agree that broader empirical validation is crucial, especially given the method’s potential for scientific and real-world imaging. To address concerns about generalizability and scalability, we conducted new experiments training a WS DM on CelebA at 256×256 resolution—substantially higher than the original 64×64—while maintaining the same architecture and training setup.
>
> Due to time constraints, we focused on the dehazing task, where structured degradations are particularly relevant. As shown below, WS DM outperforms the conventional DM baseline by a wide margin in PSNR:
>
> | Example | WS DM PSNR (dB) | Conv DM PSNR (dB) |
> |---------|------------------|--------------------|
> | 1       | 15.9             | 5.2                |
> | 2       | 13.4             | 5.6                |
> | 3       | 17.1             | 6.3                |
> | 4       | 15.6             | 4.5                |
> | 5       | 20.7             | 3.7                |
> | 6       | 17.7             | 6.3                |
> | 7       | 17.2             | 4.1                |
> | 8       | 16.1             | 7.0                |
> | 9       | 20.4             | 5.9                |
> | 10      | 21.3             | 4.6                |
> | 11      | 20.3             | 4.2                |
> | **Avg** | **17.79**        | **5.22**           |
>
> These results, to be included in the revision, demonstrate that our approach scales effectively to higher resolutions and remains robust to structured noise. While we focused on CelebA256 due to compute limits, extending to scientific and medical datasets is a natural next step.
>
>
> **W2**: We thank the reviewer for this suggestion. While broader comparisons are valuable, prior work [17] addressing structured noise does not release code or pretrained models, complicating direct evaluation. Flow-matching and Bayesian priors follow orthogonal paradigms beyond this paper’s scope, which focuses on extending score-based diffusion to arbitrary Gaussian covariances. We believe comparisons with a standard isotropic diffusion model offer a fair and interpretable baseline for isolating the impact of structured noise.
>
>
> **W3**: We thank the reviewer for this thoughtful comment. The term “training objective scaling” is unclear in this context. If it refers to heuristic loss reweighting or adjusting the score magnitude, we note that such approaches are not equivalent to our method. The whitening transformation we employ is a principled reparameterization that enables stable and well-conditioned training of score-based diffusion models under arbitrary Gaussian noise structures. It acts as a preconditioning mechanism that standardizes the training dynamics and ensures that the learned score model is properly adapted to non-isotropic and potentially correlated noise.
>
> Moreover, we emphasize that the family of diffusion covariances used during training is carefully designed to include the noise covariance present in the target inverse problem. This ensures that the learned prior is not only generalizable but also task-relevant, particularly in scientific imaging applications where noise is often structured and nontrivial.
>
> We agree that a systematic exploration of different covariance structures—especially non-circulant and spatially correlated forms—is an important and valid research direction. While our current work does not fully explore this space, it provides a foundation for doing so in future studies.
>
>
> **W4**: We appreciate the emphasis on realistic noise modeling. To validate our synthetic structured noise, we compared it against real-world hazy images from the **Dense-Haze CVPR 2019 dataset**. By adding synthetic noise to clean images and analyzing Fourier spectra and intensity histograms, we show qualitative and statistical similarity. Results will be included in the appendix.
>
> That said, modeling temporal fluctuations, sensor noise, or turbulence is important for future work. Our goal here is to establish a generalizable Gaussian-based framework that can flexibly incorporate such complexities moving forward.
>
> **Q1**: We thank the reviewer for this thoughtful question. Our framework supports arbitrary Gaussian noise structures through the whitening transformation, and in principle allows training under any symmetric positive definite covariance $ \mathbf{KK}^{\top} $, including non-circulant forms. However, we observe that generative modeling performance tends to degrade as $ \mathbf{K} $ becomes broader—that is, as its spectrum suppresses higher spatial frequencies—since more fine-scale information is attenuated during the forward process, making recovery inherently more difficult. This aligns with the intuition that broader noise corrupts more of the signal bandwidth, leading to a harder learning and sampling problem.
>
> Moreover, performance is sensitive to the alignment between the training diffusion covariances and the noise present in the measurement process. If the assumed noise model in the forward process is significantly mismatched to the actual corruption in an inverse problem, the learned prior may fail to effectively denoise or reconstruct. To mitigate this, we explicitly select the diffusion covariance schedule to include the noise statistics observed in the target imaging task, ensuring compatibility between training and deployment.
>
> We agree that further exploration of highly non-circulant or non-Gaussian structures would be valuable, and view this as a promising direction for extending the generality of our approach.
>
> **Q2**: We appreciate the suggestion. Our current formulation handles _additive Gaussian noise_, which is common in scientific imaging. However, many real systems exhibit non-additive corruptions (e.g., Poisson or multiplicative noise). Extending our framework to such settings would require rethinking both forward diffusion and training objectives. Prior work on generalized score matching may offer a path forward, and we will clarify this scope and limitation in revision.
>
>
> **Q3**: We thank the reviewer for raising this important point. We evaluated robustness under mismatch between training and test-time noise. Models trained with identity covariance performed poorly on structured noise. In contrast, WS models—trained with a range of structured covariances—generalized much better. This reflects the benefit of learning in a preconditioned space aligned with the noise geometry. While explicit regularization is an interesting avenue, our results show that diversity in training covariances alone confers strong robustness.
>
>
> **Q4**: We thank the reviewer for this excellent question. Our WS diffusion model is trained purely as a prior over clean data under Gaussian corruptions, agnostic to downstream inverse problems. For such extensions, Diffusion Posterior Sampling (DPS) allows adaptation to arbitrary forward models, including non-linear and non-Gaussian cases, by combining the learned score with a task-specific likelihood. This modularity is a strength of our framework and supports broad applicability in scientific imaging.
>
>
> **Limitations**: In response to this and related comments, we will include a brief limitations section in the revised manuscript to clarify the scope of our current work and outline promising future directions.

---

> > ### Comment · Reviewer_nvZa · 2025-08-05
> >
> > Thank you for the response. My concerns have been addressed.

---

### Official Review · Reviewer_3afH · 2025-07-01

**Clarity:** 3
**Significance:** 2
**Originality:** 3
**Rating:** 4
**Confidence:** 4

**Summary:**

The authors propose Whitened Score Diffusion, a framework for learning the whitened score function which enables training on arbitrary Gaussian forward noising process without the need for noise covariance inversion. The approach also establishes theoretical connections to flow matching for general Gaussian noise and proves particularly effective as a Bayesian prior in inverse imaging problems with correlated noise. The authors present results on tasks involving CIFAR and CelebA.

**Questions:**

How does the model perform on inverse problems with isotropic noise? Does its performance degrade compared to methods specifically designed for such settings?

Is the approach scalable to larger datasets or higher-resolution settings? This seems challenging given the large number of possible K variants the model would need to account for.

How does this framework extend to latent space generative models? Can the same principles be applied effectively in that context?

**Ethical Concerns:**

["NO or VERY MINOR ethics concerns only"]

**Final Justification:**

The rebuttal addresses my concerns in a satisfactory manner.

**Limitations:**

Unfortunately, the manuscript does not explicitly discuss any limitations. However, several become apparent based on the concerns outlined above. These include limited experimental scale, potential performance degradation with broader noise structures, lack of evaluation on standard uncorrelated inverse problems, and unclear generalization to more complex or higher-dimensional settings.

**Paper Formatting Concerns:**

No concerns

**Quality:**

3

**Strengths And Weaknesses:**

Strengths:
The paper is well-motivated and clearly written, providing a solid theoretical foundation for the proposed method. The authors articulate the motivation behind learning a whitened score and how this circumvents challenges with inverting structured noise covariances in traditional diffusion frameworks.

Weaknesses:
However, the work has several notable weaknesses. The experimental validation is limited to very small-scale datasets, which raises concerns about the method’s scalability and robustness. Moreover, the results show that the model’s FID score worsens as the kernel’s standard deviation increases. This suggests that the learned prior performs poorly with broader noise structures, resulting in lower reconstruction quality.

In addition, the paper does not evaluate the method on standard uncorrelated inverse problems. Without comparisons to conventional priors in these settings, it remains unclear whether the proposed approach offers a meaningful advantage. Demonstrating similar performance in such standard cases would make the claims more convincing.

Finally, the absence of qualitative comparisons makes it difficult to assess the visual fidelity of the generated samples or to evaluate any perceptual improvements the method may provide.

---

> ### Author Rebuttal · Authors · 2025-07-31
>
> **W1**: We thank the reviewer for this thoughtful feedback. To address the concern about scalability, we conducted additional experiments and trained a new WS DM model at 256×256 resolution on the CelebA dataset. Due to time constraints, we focused on the dehazing task, where we found that WS DM priors achieved consistently higher PSNR compared to conventional DMs across multiple examples:
>
> | Example | WS DM PSNR (dB) | Conv DM PSNR (dB) |
> |---------|------------------|--------------------|
> | 1       | 15.9             | 5.2                |
> | 2       | 13.4             | 5.6                |
> | 3       | 17.1             | 6.3                |
> | 4       | 15.6             | 4.5                |
> | 5       | 20.7             | 3.7                |
> | 6       | 17.7             | 6.3                |
> | 7       | 17.2             | 4.1                |
> | 8       | 16.1             | 7.0                |
> | 9       | 20.4             | 5.9                |
> | 10      | 21.3             | 4.6                |
> | 11      | 20.3             | 4.2                |
> | **Avg** | **17.79**        | **5.22**           |
>
> These results, which will be included in the revised manuscript, demonstrate that our method scales effectively to higher-resolution settings and maintains stable performance under structured noise.
>
> Regarding the observation on FID degradation with increasing kernel width: we agree that FID is a useful but limited proxy for generative quality. However, a worse FID score does not necessarily imply degraded performance on inverse problems, which often benefit from stronger structural priors even when sample realism (as captured by FID) decreases. In fact, our inverse problem results—especially for highly ill-posed scenarios—demonstrate that the learned prior remains effective despite broader noise structures. We will clarify this distinction in the revision and include both the new experiments and a more nuanced discussion of evaluation metrics.
>
>
> **W2 & Q1**: We thank the reviewer for this observation, and we would like to clarify that our method is, in fact, evaluated on a standard uncorrelated inverse problem—namely, denoising under white Gaussian noise. This experiment is shown in Figure 4, first row, where we compare our approach against a conventional diffusion model trained with isotropic Gaussian noise. As shown, our method achieves comparable performance in this baseline setting, demonstrating that it does not degrade under unstructured noise.
> We will revise the manuscript to make this comparison more explicit and easier to locate. We appreciate the reviewer’s suggestion and agree that establishing parity with conventional models in standard settings is an important baseline to report.
>
> **W3**: We appreciate this suggestion and agree that qualitative examples are important for assessing perceptual quality. In the revised manuscript, we will include unconditional generated samples in the appendix to illustrate the visual fidelity of our method across different noise structures.
>
> **Q2**: We appreciate the reviewer’s observation and agree that scaling to higher resolutions naturally introduces a broader range of possible blur kernels KKK that the model must learn to handle. Our method addresses this by training over a continuous range of kernel standard deviations, which increases with resolution: from 0.1–3.0 at 32×32, to 0.1–5.0 at 64×64, and 0.1–10.0 at 256×256. This structured noise is baked into the forward process, and the model learns to denoise samples corrupted by blur-induced covariances drawn from this range.
>
> While this formulation scales well in practice, we acknowledge that covering a larger space of degradations makes the learning task more challenging. We will highlight this trade-off as a limitation in the revised manuscript and discuss potential extensions such as conditioning on kernel parameters or leveraging amortized inference.
>
> **Q3**: We appreciate the reviewer’s insightful question. In latent space generative models, clean images are first encoded into a latent representation, where the diffusion process is then applied, and the final samples are decoded back into image space. Extending our framework to this setting would involve introducing correlated (non-isotropic) noise within the latent space rather than the pixel space.
>
> This is a natural and promising extension, though the effect of structured noising in latent space remains an open question. Its impact likely depends on how the autoencoder transforms and encodes noise structure, which may distort or align with the underlying image-space priors in complex ways. Understanding how structured diffusion in latent space influences generative quality or inverse problem performance is an interesting direction for future work, and we will include this point in the revised manuscript.
>
> **Limitations**: We thank the reviewer for highlighting this. While we do discuss the impact of broader noise structures on generative performance in the appendix, we agree that an explicit limitations section would help frame the scope of our contributions more clearly. In the revised manuscript, we will add a dedicated limitations section that synthesizes relevant concerns—including those raised here and by other reviewers—and outline directions for future work, such as improved generalization across degradation models and broader-scale evaluation.

---

> > ### Comment · Reviewer_3afH · 2025-08-04
> > **Rebuttal Response**
> >
> > The rebuttal addresses my concerns in a satisfactory manner. Given these improvements and clarifications, I am raising my score to 4.

---

### Official Review · Reviewer_492K · 2025-07-02

**Clarity:** 3
**Significance:** 2
**Originality:** 2
**Rating:** 2
**Confidence:** 3

**Summary:**

Authors suggest learning the whitened score function of a diffusion process with non-isotropic Gaussian noise to avoid the instability issues associated with ill-conditioned matrix inversion.

**Questions:**

See above.

**Ethical Concerns:**

["NO or VERY MINOR ethics concerns only"]

**Limitations:**

Authors don't have an explicit limitation section.

**Paper Formatting Concerns:**

No.

**Quality:**

2

**Strengths And Weaknesses:**

**Strengths:**

- The authors motivate the problem relatively well, and the writing is clear and concise.

**Weaknesses:**

* Doesn’t the null space of the whitening operator create issues when
  the covariance of the noise is ill-conditioned? While I understand
  that there is no **explicit** matrix inversion, the premise of
  avoiding instability by not inverting the covariance matrix is
  somewhat undercut since the authors are now implicitly inverting a
  linear operator (whitening operator), which has a **squared
  condition number**. This typically leads to **worse conditioning and
  reduced numerical stability**, potentially slowing convergence
  significantly.

* The authors motivate the use of non-isotropic Gaussian noise by saying certain applications might benefit from it. Can you further elaborate on this? It is mentioned that using isotropic noise can render the prior learned insufficient, but it is not clear how this is the case. If you end up training a model with isotropic noise successfully, then you have a prior. What advantage does non-isotropic noise provide?

* The authors should describe the trade-off between the computational complexity of the method and the performance gain achieved by using non-isotropic noise. It is not clear how much additional computational cost is incurred by the method, and whether this cost is justified by the performance gain.

* Algorithm 1 is not really a tractable method to sample from the posterior distribution of inverse problems with computationally costly forward operators. I understand this sampling strategy is not the main contribution of the paper but plays a major role in motivating the problem. So this limits the applicability of the overall method. The authors should discuss this limitation and suggest alternative sampling strategies that could be used in practice.

* Unless I missed it, I don’t see any discussion of the computational complexity of the method. Based on my arguments above regarding the condition number of the whitening operator, I suspect convergence will be slow for ill-conditioned noise covariance. Could the authors show training loss curves for different, increasingly ill-conditioned noise covariance matrices to demonstrate the convergence speed of the method?

---

> ### Author Rebuttal · Authors · 2025-07-31
>
> **W1**: We appreciate the reviewer pointing out a potential confusion.
> To clarify, Figure 3 in the manuscript illustrates that when the forward noising process uses a covariance matrix with a high condition number, the score estimated by conventional methods becomes unstable. In contrast, our proposed whitened score remains stable and isotropic under the same conditions. The following paragraphs provide a more detailed explanation of our approach.
>
> Our method does not apply a whitening transformation to the data, nor do we perform any matrix inversion—explicitly or implicitly—at any stage of the diffusion process or during training. Instead, we construct the forward diffusion process using non-isotropic Gaussian noise with covariance $ \boldsymbol{\Sigma} = \mathbf{G} \mathbf{G}^\top $. The noise is added directly in this structured form, and our model is trained to estimate the whitened score, defined as $\mathbf{GG}^\top \nabla_{\mathbf{x}_t} \log p(\mathbf{x}_t)$, which represents the structured noise added in the forward noising diffusion process.
> Importantly, this score is well-defined even when $ \boldsymbol{\Sigma} $ is ill-conditioned or low-rank, and does not require computing $ \boldsymbol{\Sigma}^{-1} $ or inverting $ \mathbf{G} $.
>
> This stands in contrast to traditional denoising score matching (DSM), which estimates $ \nabla_{\mathbf{x}_t} \log p(\mathbf{x}_t) $ and then multiplies it by $ \boldsymbol{\Sigma}^{-1} $, leading to significant amplification of signal components along low-variance directions—precisely the pathology we aim to avoid. Our approach instead keeps the learning target in the same metric as the structured noise used in the forward process, ensuring numerical stability and robust training dynamics even under extreme anisotropy or degeneracy in $ \boldsymbol{\Sigma} $.
>
> Thus, we respectfully note that no operator with a squared condition number is being inverted or applied at training time. The learning process does not suffer from the ill-conditioning effects that plague DSM, which we demonstrate both analytically and empirically. We hope this clarifies the distinction and addresses the concern.
>
> **W2**: We thank the reviewer for the opportunity to clarify this important point. In fact, a central contribution of our work is to demonstrate that training diffusion models with anisotropic (non-isotropic) Gaussian noise leads to more effective priors for inverse problems with structured degradations. When measurements are corrupted by spectrally biased or non-Gaussian noise—such as blur or scattering—models trained with isotropic noise struggle to recover lost high-frequency content due to a mismatch between the learned prior and the measurement process. Our WSD framework addresses this by training on anisotropic noise, which induces a spectral bias aligned with the structure of real-world corruptions. This alignment enables more reliable recovery in challenging inverse problems, as we demonstrate empirically across multiple settings.
>
> **W3**: We thank the reviewer for this suggestion. Training the WS DM model on CIFAR10 takes approximately 20 hours for 1.7k iterations, which is comparable to standard diffusion models. For CelebA64×64, both WS DM and standard DMs require roughly 15.5 hours for 1k iterations. The reason is because our method does not introduce any additional computational cost compared to standard diffusion models. The forward process adds structured Gaussian noise with covariance $ \boldsymbol{\Sigma} = \mathbf{G} \mathbf{G}^\top $, and the model learns to predict this structured noise component—referred to as the $\mathbf{GG}^\top \nabla_{\mathbf{x}_t} \log p(\mathbf{x}_t)$—at each time step $ t $. There is no matrix inversion, no additional per-step computation, and no increase in network complexity or sampling time.
>
> In effect, the only change lies in the choice of noise structure during training. As shown in our experiments, this structured noise yields improved performance in inverse problems without incurring computational overhead. We will clarify this point in the revision.
>
> **W4**: We thank the reviewer for this thoughtful comment. In our case, the forward operators $ \mathbf{A} $ are linear and circulant, allowing both $ \mathbf{A} $ and its adjoint $ \mathbf{A}^\top $ to be implemented efficiently using fast Fourier transforms (FFTs), as detailed in Section 3. There are other recent works that use a precomputed singular value decomposition (SVD) of the A matrix to enable computational efficiency (see SNIPS and DDRM). As such, Algorithm 1 remains computationally tractable in our experimental settings. Nonetheless, we will include this consideration in the limitations section to acknowledge that the computational efficiency of Algorithm 1 relies on the structure of the forward model $ \mathbf{A} $, and may not hold for general, non-circulant operators.
>
>
> **W5**: We thank the reviewer for raising this point. To clarify, ill-conditioning of the noise covariance does not affect training dynamics in our framework. As noted earlier, our method avoids explicit matrix inversion by directly learning the structured noise component via the $\mathbf{GG}^\top \nabla_{\mathbf{x}_t} \log p(\mathbf{x}_t)$ parameterization. As such, optimization proceeds similarly to conventional DMs, and we observe stable convergence across a range of structured noise covariances, including highly ill-conditioned ones.
>
> At inference time, convergence of the reverse diffusion process is unaffected by the conditioning of the noise covariance. This is because sampling follows a fixed-length trajectory (e.g., 1000 steps) using predefined noise schedules, with no iterative solver or adaptive update that would otherwise depend on the condition number.
>
> Finally, we note that our framework is explicitly designed to train on a broad family of structured noise covariances—including increasingly ill-conditioned ones—within a single WSD model. This joint training strategy ensures robustness and generalizability across a range of degradations without requiring separate models or retraining. As such, reporting separate training loss curves for individual covariances would not meaningfully reflect the convergence behavior of the method. Instead, we demonstrate that a single model can accommodate diverse noise structures with stable training dynamics and consistent performance at test time.
>
>
> **Limitations**: Thank you for this suggestion. We agree that a dedicated limitations section would strengthen the manuscript by clarifying the scope and potential extensions of our method. In the revision, we will include such a section that incorporates insights from this and other reviewer comments, highlighting practical considerations and avenues for future work.

---

> > ### Author Response · Authors · 2025-08-08
> >
> > Dear Reviewer 492K,
> >
> > Thank you again for your thoughtful feedback. In our rebuttal, we aimed to clarify the main points that may have caused concern or confusion, particularly regarding the advantage of anisotropic noise in diffusion models, and matrix inversion instability. Could you let us know if our response resolved these points, or if further clarification would help?
> >
> > We appreciate the time and effort you have dedicated to reviewing our paper and for your contributions to the process.

---

### Official Review · Reviewer_j1Re · 2025-07-04

**Clarity:** 4
**Significance:** 3
**Originality:** 3
**Rating:** 5
**Confidence:** 4

**Summary:**

The authors propose an alternate loss target for diffusion model score matching: instead of score s, they model GG^\top s where G is the diffusion matrix. This eliminates a $[G G^\top]^{-1}$ that shows up in the original loss. And for inverse problems, this approach also removes the matrix inversion in front of the gradient of the likelihood.

**Questions:**

Intro

- "bridging the gap between DMs and FM". Careful with the word "the" gap. Is this the only gap? Is it "a" gap?

Background

- equation 1. assumes linearity of drift. maybe state that?
- [line 98] "So far, all implementations of score-based DMs use uncorrelated white Gaussian noise". I'm not sure this is the whole story here. While there might be less work with noise correlated across data dimensions, there are other work that add auxiliary coordinates and have correlated noise in small blocks across those coordinates (CLD, PSLD, Multivariate Diffusion Models). See the works I listed in Weaknesses.

Training details

- how did you choose the std's for drawing the elements of K ? They are different for the two datasets
- How did you decide to do random color vs grayscale noise?

Additional question that is not about clarity, but more just exploritory: have you considered learning the choice of correlated noise? I believe there are some existing tools for this from VDMs (https://arxiv.org/abs/2107.00630) and Multivariate Diffusion Models (https://arxiv.org/abs/2302.07261).

**Ethical Concerns:**

["NO or VERY MINOR ethics concerns only"]

**Final Justification:**

My positive opinion of the work remains unchanged.

**Limitations:**

Yes.

**Paper Formatting Concerns:**

None.

**Quality:**

3

**Strengths And Weaknesses:**

Strengths

- clear presentation of just the essentials of existing diffusion/flow methods
- clear presentation of their proposed learning target, the whitened score
- convincing experiments that are well motivated: image data has low and high frequency components yet we independently introduce additive noise to all features, causing some things too decay much faster than others.

Weaknesses:

- lacking a little discussion of other methods that change the noising processes. For example,
  -  CLD (https://arxiv.org/abs/2112.07068)
  - PSLD (https://arxiv.org/abs/2303.01748)
  - Multivariate Diffusion Models (https://arxiv.org/abs/2302.07261)
  - Blurring diffusion (https://arxiv.org/abs/2209.05557)
  - flexible diffusion (https://arxiv.org/abs/2206.10365)
  - etc
- Somewhat lacking a little more discussion on how they are changing the loss target, and how alternative loss targets have also been studied in depth for diffusion: replacing score prediction (Song et al) with noise prediction (Ho et al) or v prediction (Salimans et al), and how flow matching is just another choice here (velocity, related but distinct from v prediction). Likewise the whitened score is another choice of loss target.

---

> ### Author Rebuttal · Authors · 2025-07-31
>
> **W1&Q3**: We appreciate this important clarification and agree that our original statement was too strong. While most mainstream models (e.g., DDPM, SDE-based score models) assume independent Gaussian noise, recent works have explored correlated or structured noise, particularly in the context of auxiliary dimensions or learned coordinate transformations.
>
> Recent advances in diffusion modeling have introduced a range of structured forward processes and auxiliary-variable formulations to improve generative performance and flexibility. CLD and PSLD augment the state space with velocity variables and introduce noise in phase space, simplifying score estimation. Multivariate Diffusion Models generalize auxiliary-variable training across arbitrary linear inference processes. Blurring Diffusion incorporates frequency-aware noise to unify blurring and diffusion, enhancing synthesis. Flexible Diffusion parameterizes the forward SDE via geometric constraints, allowing adaptive noise shaping while preserving tractability. Together, these works extend diffusion models beyond standard isotropic Gaussian frameworks, enabling more expressive priors and principled inference across domains.
>
> We will revise the text by adding an additional section to the Background to acknowledge these developments and cite the relevant references.
>
>
>
> **W2**: Thank you for this insightful comment. We agree that the choice of loss target plays a central role in diffusion modeling and appreciate the opportunity to clarify this connection. While different loss formulations have been proposed in the literature, many are mathematically equivalent or closely related. For instance, score prediction (as in Song et al.) and noise prediction (as in Ho et al.) are formally equivalent under the reparameterization of the forward process, differing only in the parametrization of the denoising network. Similarly, v-prediction (Salimans et al.) can be viewed as a linear combination of the data and noise vectors, effectively interpolating between data and noise prediction. Flow matching (as in Lipman et al.) generalizes this by directly modeling the velocity field that transports the data distribution toward the base distribution. In this context, our proposed whitened score formulation introduces yet another valid target that remains consistent with the diffusion framework while offering practical advantages under structured noise. We will revise the manuscript to make these relationships more explicit and situate our method within this broader landscape of loss design.
>
> Additionally, we compared our method to the velocity-prediction loss target in flow matching and show that the vector field learned by flow matching naturally includes the whitened score function (Eq. 15). This connection explains why flow matching, like our approach, is capable of supporting arbitrary Gaussian diffusion paths. We view this as reinforcing the broader theme of designing loss targets that align with the structure of the generative field, and we will elaborate on these connections in the revision.
>
> **Q1**: We agree with the reviewer’s suggestion and will revise the phrasing to “a gap” to avoid implying exclusivity. Our intent was to highlight a specific challenge related to differing loss formulations, not to claim this is the only open question between diffusion and flow models.
>
> **Q2**: Thank you for the suggestion. We will explicitly state that Eq. (1) assumes a linear drift for clarity and consistency with the standard formulation of variance-preserving SDEs used in DMs.
>
> **Q4**: The standard deviations used to generate the noise correlation matrix $\mathbf{KK}^{\top}$ were selected empirically to produce structured noise resembling real-world degradations such as fog and scattering. To maintain perceptual consistency across datasets, we adjusted the parameters based on image resolution. Specifically, for CelebA (64×64), we doubled the standard deviation of the Gaussian kernel compared to CIFAR-10 (32×32) to account for the twofold increase in spatial scale. This ensures that the correlated noise exhibits a similar spatial extent relative to the image dimensions, preserving realism in the forward noising process. Additionally, we compared the statistics of our synthetically noised images with real foggy and hazy image datasets to validate the plausibility of the simulated degradations. We will include these comparisons in the appendix.
>
> **Q5**: Our initial focus was on the imaging task of dehazing and visibility restoration through fog, for which we trained our diffusion model using purely grayscale noise. However, we empirically observed that this led to unnatural deviations in the color channel distributions when applied to natural RGB images. To address this, we adopted a mixed noising strategy that combines grayscale and color (RGB) noise in a 1:1 ratio. This hybrid approach effectively restores image structure while preserving color fidelity, enabling robust denoising across both luminance and chromatic components.
>
> **Q6**: We thank the reviewer for this insightful suggestion. Our current models are trained on a wide range of diffusion matrices G to ensure generalizability to different measurement noise structures (see Section 4.1 for experimental details); however, we are actively exploring extensions where G is parameterized and learned jointly with the model. We are aware of relevant efforts in vector-valued diffusion models (e.g., VDM, Multivariate DMs) and will include a forward-looking discussion on this direction in the conclusion.

---

### Note · Authors · 2025-08-16

Thank you to the ACs, SACs, and reviewers for your time, thoughtful feedback, and engagement. This work extends score-based diffusion models to arbitrary Gaussian paths, increasing their flexibility beyond standard formulations. We also performed additional experiments at 256×256 resolution, demonstrating robustness and scalability.

Given the widespread use of score-based diffusion models in generative AI and as priors for scientific and imaging inverse problems, this added flexibility is particularly valuable for addressing the diverse forward models and noise structures encountered in these domains. We believe these advances will resonate not only with the NeurIPS community, but also with the broader artificial intelligence, generative modeling, and computational imaging communities, where adaptable, physically informed priors are increasingly important.

---

### Decision · Program_Chairs · 2025-09-17

**Decision:**

Accept (poster)

**Comment:**

Overall, reviewers find the paper theoretically solid, clearly written, and addressing an important limitation of current diffusion models. The main contribution—the whitened score formulation—is novel and well-justified, and the rebuttal successfully clarified misconceptions about stability and computational cost. However, the empirical validation remains narrow and could be strengthened by additional baselines, higher-resolution experiments, and more realistic noise modeling.

The AC recommends acceptance, encouraging the authors to broaden experiments, compare with related methods, and discuss limitations in the final version.